# Translatome profiling in fatal familial insomnia implicates TOR signaling in somatostatin neurons

Susanne Bauer[1], Lars Dittrich[2], Lech Kaczmarczyk[1,2], Melvin Schleif[2], Rui Benfeitas[3], Walker S Jackson[1,2]

**Selective neuronal vulnerability is common in neurodegenerative diseases but poorly understood. In genetic prion diseases, including fatal familial insomnia (FFI) and Creutzfeldt–Jakob disease (CJD), different mutations in the *Prnp* gene manifest as clinically and neuropathologically distinct diseases. Here we report with electroencephalography studies that theta waves are mildly increased in 21 mo old knock-in mice modeling FFI and CJD and that sleep is mildy affected in FFI mice. To define affected cell types, we analyzed cell type–specific translatomes from six neuron types of 9 mo old FFI and CJD mice. Somatostatin (SST) neurons responded the strongest in both diseases, with unexpectedly high overlap in genes and pathways. Functional analyses revealed up-regulation of neurodegenerative disease pathways and ribosome and mitochondria biogenesis, and down-regulation of synaptic function and small GTPase-mediated signaling in FFI, implicating down-regulation of mTOR signaling as the root of these changes. In contrast, responses in glutamatergic cerebellar neurons were disease-specific. The high similarity in SST neurons of FFI and CJD mice suggests that a common therapy may be beneficial for multiple genetic prion diseases.**

## Introduction

Neurodegenerative diseases (NDs) are widely thought to be caused by the misfolding of specific proteins. They tend to emerge in middle to late life and slowly, progressively destroy the brain. A striking feature of NDs is the selective vulnerability of specific neurons and brain regions in early disease stages, which occurs despite widespread expression of the disease-causing protein.

Selective vulnerability is particularly curious in the case of genetic prion diseases, where different point mutations in the ubiquitous prion protein (PrP) have been linked to different diseases affecting different brain regions and manifesting with distinct neuropathological hallmarks and clinical signs (1). Genetic Creutzfeldt–Jakob disease (hereafter Creutzfeldt–Jakob disease

[CJD], although it differs from non-genetic forms) can be caused by several mutations but is most commonly linked to the E200K substitution (2). Clinical signs include rapidly progressing dementia, balance and gait disturbances, myoclonus, and sometimes seizures. Neuropathological hallmarks of CJD include spongiform degeneration accompanied by astrogliosis and neuronal loss in the cortex, deposition of PrP aggregates that resist proteinase K digestion (PrP$^{res}$), and mild spongiform degeneration in the molecular layer of the cerebellum (3, 4). The most common genetic prion disease is fatal familial insomnia (FFI), caused by a D178N substitution (5). This devastating disease typically begins with rapidly progressing insomnia, autonomic, and motor disturbances, followed by cognitive decline (4, 6). Neuronal loss is most severe in the anterior and medial dorsal thalamus and accompanied by astrogliosis. However, in contrast to most other prion diseases, spongiform degeneration and PrP$^{res}$ are typically absent and usually occur only in cases with a prolonged disease course (7). Cerebellar neuropathology includes gross atrophy (5), prominent loss of Purkinje cells and morphological changes to granular neurons (4). Remarkably, these inherited diseases usually emerge in middle to late life, the same age as other causes of prion diseases (PrDs) even though the mutant protein is expressed throughout life.

Clinical signs of FFI and CJD correlate with loss of function of affected brain regions but the underlying mechanisms for these disease-specific patterns are unknown. In addition to regional vulnerability, certain cell types are highly vulnerable. Thus, rather than analyzing brain regions, an alternative approach is to analyze cell type–specific responses in the presence of mutated PrP. This can be accomplished by analyzing the translatome, mRNAs associated with ribosomes, from specific cell types. In this study, we analyzed translatome changes in vGluT2$^+$ glutamatergic neurons (excitatory) and Gad2$^+$ GABAergic (generally inhibitory) neurons in the cerebrum and cerebellum, as well as cerebral GABAergic subpopulations expressing neuropeptides parvalbumin (PV) or somatostatin (SST), which are non-overlapping in most brain regions. These are of particular interest for our study because PV-expressing neurons in the cortex or cerebellum are highly vulnerable in PrDs (8, 9). Moreover, PV$^+$ (10) and SST$^+$ neurons (11, 12) are highly vulnerable in other NDs and psychiatric disorders (13).

[1]Department of Biomedical and Clinical Sciences, Wallenberg Center for Molecular Medicine, Linköping University, Linköping, Sweden  [2]German Center for Neurodegenerative Diseases, Bonn, Germany  [3]Department of Biochemistry and Biophysics, National Bioinformatics Infrastructure Sweden (NBIS), Science for Life Laboratory, Stockholm University, Stockholm, Sweden

Correspondence: walker.jackson@liu.se

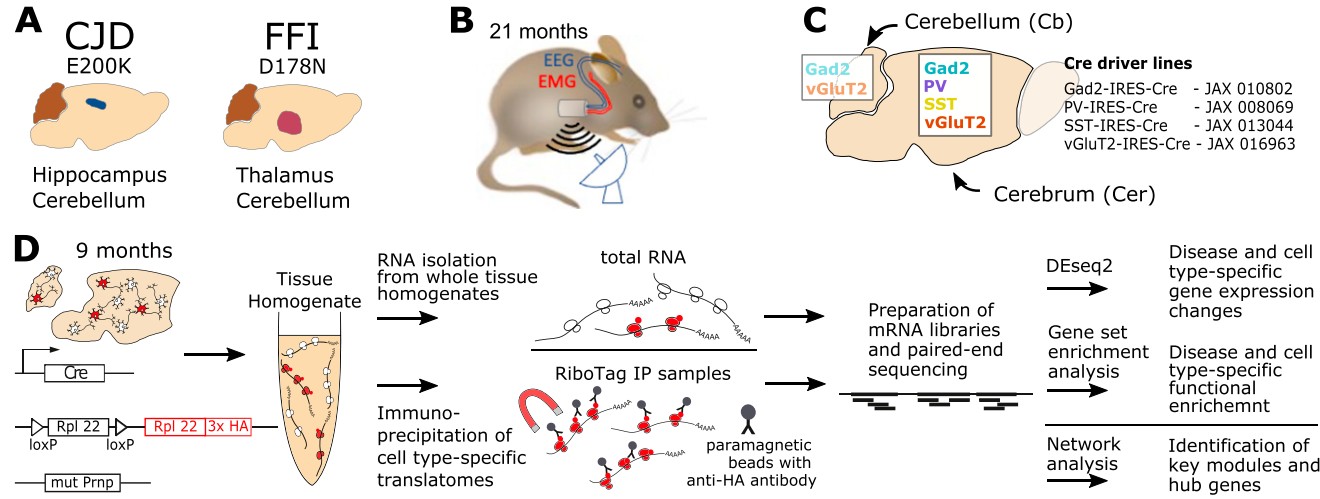

**Figure 1. Experimental setup and workflow.**
**(A)** This study compared two *Prnp* knock-in mouse models of genetic prion diseases, Creutzfeldt Jakob disease (CJD) and fatal familial insomnia (FFI), relative to age-matched controls expressing wild-type *Prnp*. Mutant *Prnp* mice show selective vulnerability in the hippocampus (blue, CJD) and thalamus (red, FFI), respectively, in addition to secondary pathology in the cerebellum (brown). **(B)** Telemetric electroencephalography (EEG) and electromyography (EMG) were performed at 21 mo of age to characterize the sleep phenotype. **(C)** For cell type–specific translatome analysis, four Cre-driver lines were used to target neuronal subtypes in the cerebellum and cerebrum. Gad2: Glutamate decarboxylase 2 marks GABAergic (gamma-aminobutyric acid) neurons; vGluT2: vesicular Glutamate Transporter 2 marks glutamatergic neurons; PV (parvalbumin) and SST (somatostatin) target subtypes of GABAergic neurons that are typically non-overlapping. vGluT2 and Gad2 expressing neurons were analyzed separately in the cerebellum (orange, light blue) and cerebrum without olfactory bulb (red, turquois). SST (yellow) and PV (purple) expressing neurons were analyzed only in the cerebrum. **(D)** Schematic workflow for preparation and analysis. Mice double homozygous for mutated or unmodified *Prnp* and RiboTag were crossed with homozygous Cre driver lines to obtain expression in the desired cell types. After euthanasia of mice at 9 mo (average age: 9.3 mo, SD: 0.7), RiboTag samples were obtained from cerebrum or cerebellum by immunoprecipitation of HA-tagged ribosomes with anti-HA antibody-bound magnetic beads. For a subset of biological replicates, total RNA was prepared from tissue homogenate as input control. After library preparation and sequencing, differential gene expression and functional analyses were performed at a disease and cell type level. To identify new candidate genes, we constructed and analyzed a weighted co-expression network for SST neurons. HA, hemagglutinin; SD, standard deviation.

The cerebellum was also of interest because it is affected in both diseases. To obtain cell type–specific translatome profiles, we used RiboTag (14) directed to specific neuronal populations in knock-in mouse models of FFI and CJD at a pre-symptomatic stage. This experimental design resulted in several unexpected findings. Despite the vastly different regional pathologies, cell type specific responses were similar between FFI and CJD but were strikingly different from those in a model of acquired prion disease in which PrP was expressed from the same genetic locus. Also, we describe for the first time the extensive changes in SST+ neurons at a pre-symptomatic disease stage, a cell type that has hitherto been understudied in PrDs.

## Results

We previously developed knock-in mouse models of genetic CJD and FFI linked to E200K and D178N mutations, in the endogenous mouse *Prnp* gene (15, 16). These models developed late onset, progressive diseases that replicate several key pathological features of the respective human diseases, and importantly, differ from each other in pathological changes and affected brain regions. FFI mice experience neuronal loss and reactive astrocytosis in the thalamus and atrophied cerebellum (15). In contrast, CJD mice develop PrP^res and spongiosis, hallmarks of the human disease, most prominently in the hippocampus, and PrP^res in the molecular layer of the cerebellum (16) (Fig 1A). PrP in CJD mice had a slightly

altered glycoform pattern, suggesting a slightly altered path through the secretory system. In contrast, in FFI mice, mono- and unglycosylated PrP were nearly absent and the total amount of all forms was only 25% of normal levels (15), suggesting the FFI mutant is subjected to intensive quality controls and that the mammalian brain responds to these mutant proteins differently. Automated mouse behavioral analysis used to measure multiple activities of mice (e.g., roaming, grooming, distance traveled, and rest) in home cages (17) indicated sleep was fragmented and core body temperature measurements suggested FFI mice had impaired sleep regulation at 16 mo of age (15), but electroencephalography (EEG) measurements were not attempted then because of biosafety constraints. Consideration of the neuropathological changes and in vivo clinical abnormalities measured by automated mouse behavioral analysis and in vivo magnetic resonance imaging led to the general picture that disease emerged at ~16 mo of age for both models (15, 16).

### Neural activity is mildly affected in old FFI and CJD mice

To rigorously characterize the general sleep features and neural health in these models, we used the same EEG methods (Fig 1B) we applied previously to the RML (Rocky Mountain Labs) model of acquired PrD (18). Because a telemetric recording system was used, mice could roam freely in their cage, thereby avoiding artifacts from tethering. In that study, θ frequency waves increased as disease progressed, like observations in several human PrDs (19). Notably,

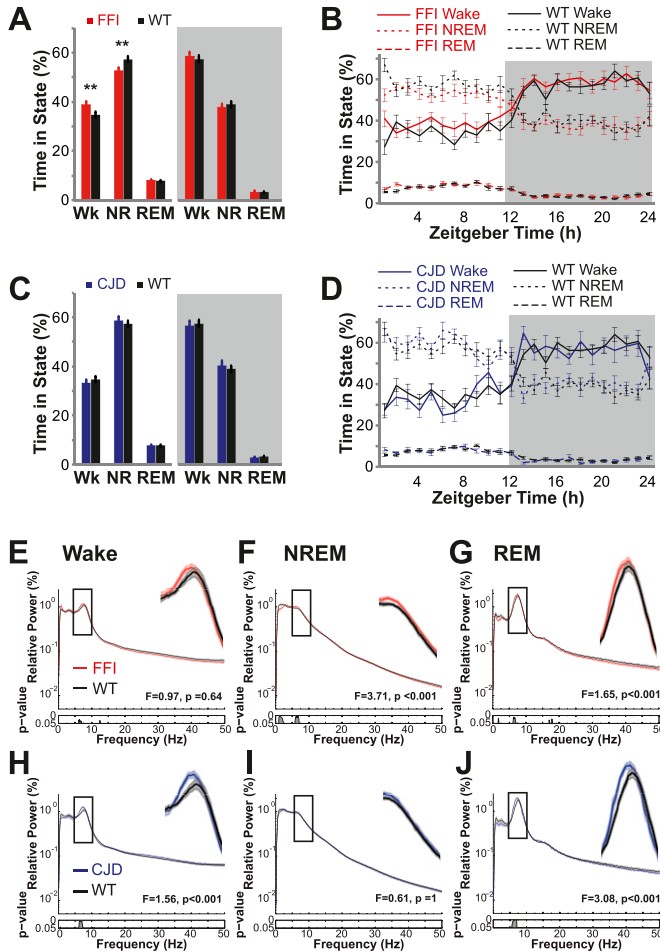

**Figure 2. Aged fatal familial insomnia (FFI) and Creutzfeldt–Jakob disease (CJD) mice have mild EEG abnormalities.**
**(A)** Baseline recordings covering 24 h were divided into 10-s bins and scored as Wake (Wk), NREM (NR), or REM. Analyzing data from the complete dark phase (grey background) or light phase (blank background) revealed that, compared with WT mice (n = 14), FFI mice (n = 15) had slightly increased wake and decreased NREM sleep during the light phase (**, t test, P < 0.01). **(B)** The same data divided into 1 h bins beginning when lights turn on (Zeitgeber Time 0); the grey background represents the dark cycle. The sleep states in FFI mice (red lines) generally matched the sleep states in WT mice (black lines) across all bins. **(C, D)** The same analytical procedures applied to CJD mice (n = 8, blue) show no changes to sleep. **(E, F, G, H, I, J)** Power-frequency spectra depict, for each wave frequency (x-axis), the proportion of power (y-axis) that it contributes to the full spectrum measured (0–50 Hz). Rectangles mark the theta band frequencies (5–10 Hz) which is magnified to the right. Below each panel is a P-value continuum using the same x-axis scale as the power-frequency spectra. **(F, G, H, J)** The panels show theta is increased in FFI during NREM (F) and REM (G) sleep and increased in CJD mice in Wake (H) and REM (J) sleep. NREM, non-rapid eye movement; REM, rapid eye movement.

sleep was not affected in RML mice, even in late stages (18). Because we previously observed that behavioral activity in FFI mice was only mildly affected at 16 mo of age, which is likely a result of only mildly diminished neural health at that time point, we sought to increase the possibility of detecting EEG abnormalities by studying old mice at ~21 mo of age (mean = 20.8, SD = 2.3). Surprisingly, considering that in FFI mice temperature was dysregulated and that sleep bouts

were disrupted according to an automated video-based system (15), sleep was only modestly affected (Fig 2A). During lights on, when mice sleep the most, non-rapid eye movement (NREM) sleep was reduced (P < 0.01), and wake was increased (P < 0.01), with no difference during lights off (Fig 2A). Examining the data in 1-h bins showed there was not a specific time when sleep loss occurred (Fig 2B). To test if sleep control was vulnerable to external manipulation, we measured the response to 6 h of sleep deprivation, which showed no significant differences between FFI and control mice (Fig S1A and B). Notably, some mice assigned to this study died without being recorded, which may have selectively removed mice with the most clinically advanced disease (details in the Materials and Methods section). Considering this potential bias, and that sleep abnormalities are sometimes absent in humans with FFI (5, 20), the small effect detected in this study is not surprising. CJD mice studied in parallel showed no abnormality in baseline sleep (Fig 2C and D) or in response to sleep deprivation (Fig S1C and D). Interestingly, θ frequency waves were increased in FFI mice during NREM and REM sleep (Fig 2F and G), and in CJD mice during wake and REM sleep (Fig 2H and J), mimicking this potential biomarker of human PrD (19). The lack of θ increase during wake in FFI (Fig 2E) and NREM in CJD (Fig 2I) may be a reflection of differential selective vulnerability between the diseases. Therefore, despite these models showing neuropathological and behavioral changes by this age, there are only mild changes to θ frequency, sleep is only mildly disrupted in FFI mice, and the overall picture that disease begins at ~16 mo is unchanged.

In our recent study on RML-infected mice mentioned above (18), we found that before EEG, behavioral or neuropathological changes emerged, RiboTag profiling identified specific cell types with altered translatomes. To study a similar disease stage as done for that study (56% of disease onset), these RiboTag experiments included WT, FFI and CJD mice at 9 mo of age.

## Capture of cell type–specific translatome with RiboTag immunoprecipitation (IP)

Because the total composition of proteins (referred to as the proteome) is better reflected by the total composition of mRNAs undergoing translation (known as the translatome) than total RNAs (the transcriptome) (21), we sought to study the translatome of specific cell types with RiboTag mice (14). The RiboTag transgene is embedded into the gene encoding the large subunit ribosomal protein 22 (Rpl22) whereby, following activation with a cell type–specific Cre recombinase, a version of Rpl22 fused to the hemagglutinin (HA) antibody epitope is expressed, and HA-tagged ribosomes can be immunoprecipitated (Fig 1D). As a part of the large ribosomal subunit, Rpl22 will only associate with mRNAs when part of a complete, functional ribosome and thus RiboTag-captured mRNAs represent the translatome. Importantly, mRNAs associated with only the small subunit, as well as those not associated with ribosomes at all, are not captured by RiboTag and are therefore excluded from our analysis.

Driver lines expressing Cre directed by the genes encoding Gad2 (22), vGluT2 (23), PV (24), and SST (22) were used to achieve cell type–specific expression of the RiboTag transgene. This enabled us to target wider populations of glutamatergic and GABAergic neurons, as well as PV+ and SST+ GABAergic subtypes. Using a selection

of cell type marker genes, we recently confirmed by both immunofluorescence and RNA-seq of RiboTag IPs that these Cre lines lead to specific and selective activation of RiboTag expression [18]. The study group in the current report was age-matched (mean = 9.3 mo, SD = 0.7, details of ages in Table S1), double heterozygous for RiboTag and Cre, and homozygous for either FFI, CJD, or WT (unmodified wild-type) *Prnp* alleles (Fig S2A). Because the commonly used C57Bl/6 strain is hyperactive at night [25], and we worried this would introduce unwanted changes to gene expression patterns, we used the calmer 129S4 strain for all mice in this report (details in the Materials and Methods section). Because the cerebellum was affected in both FFI and CJD models, and the remaining part of the brain (hereafter cerebrum) had distinct brain regions that were targeted in each model, the cerebellum and cerebrum from each brain were studied separately (Fig 1C). RiboTag IPs (Fig 1D) were prepared for all cell types for cerebrum samples, but only for Gad2 and vGluT2 for cerebellum, because in the cerebellum PV-Cre induces RiboTag expression in the same cells as Gad2-Cre, whereas SST-Cre induces RiboTag expression in very few cerebellar cells. Consequently, we profiled six cell types, encompassing two brain regions, in two genetic PrDs. To verify the isolation of cell type specific translatomes in RiboTag IP samples, we established a reference by analyzing total mRNA obtained from tissue homogenates before RiboTag IPs for a subset of biological replicates (Fig 1D). As was expected, yields varied greatly based on the abundance of the targeted neurons. Average yields from RiboTag IPs ranged from 76 ng of RNA from SST samples (least abundant cell type; SD: 25 ng) and 910 ng from cerebral vGluT2 samples (most abundant cell type; SD = 230 ng). However, we found no significant differences in RiboTag IP RNA yields between different genotypes (Fig S2B).

After sequencing of RNA and mapping of reads, we detected on average 12,560 expressed genes in RiboTag IP samples. Although the average number of detected genes varied slightly by cell type, we did not find a significant difference in detected genes between genotypes in the same cell type (Fig 3A). As expected, principal component analysis (PCA) showed differences between total mRNA samples based on the region (cerebellum versus cerebrum) but not cell types (Fig 3B). In contrast, IP samples showed clear differences based on regions and cell types (Fig 3C). This was also apparent through comparisons of expression of cell type marker genes in RiboTag IP samples normalized to total RNA expression levels, which revealed the expected relative enrichment of general GABAergic and glutamatergic neuronal marker genes in respective RiboTag IP samples (Fig 3D and E). Targeting of specific subclasses of GABAergic neurons was confirmed by up-regulation of PV- or SST-specific marker genes in the respective samples, whereas *Htr3a* (serotonin receptor 3A) and *Vip* (vasoactive intestinal peptide), GABAergic markers absent from SST and PV neurons, showed the predicted enrichment in Gad2[+] and depletion in PV[+] and SST[+] IPs (Fig 3D). In the cerebellum, Gad2[+] IPs were enriched for marker genes of several cerebellar GABAergic cell types such as Purkinje, basket, Golgi, and stellate cells, whereas vGluT2[+] IPs showed enrichment for granule cell markers (Fig 3E) [26, 27]. As expected, astrocyte and microglia marker genes [28] were depleted in all IP samples. These results indicate that cell type–

specific translating mRNA was successfully isolated from the intended neuronal subpopulations.

## *Prnp* expression varies with cell type and sequence

One potential explanation for selective vulnerability is that vulnerable cell types express high levels of toxic protein. To investigate this possibility, we examined the expression levels of *Prnp* in the targeted cell types based on transcript per million (TPM) (Fig 3F). Unexpectedly, *Prnp* was expressed almost twofold higher in vGluT2[+] neurons than in GABAergic cell types. These differences were detected in all three genotypes. Higher *Prnp* expression in vGluT2 neurons may partially explain the selective vulnerability in these models because the regions most affected, thalamus and hippocampus, are predominately glutamatergic and have high *Prnp* expression, second only to the cortex [29]. This analysis also showed that FFI mice had slightly lower *Prnp* expression. This tendency was most pronounced in glutamatergic neurons and only significant in cerebral vGluT2[+] neurons (Kruskal–Wallis, $P$ = 0.026, $\chi^2$ = 7.312). This observation is consistent with the reduced PrP levels previously reported in FFI mouse brains, suggesting that the D178N mutant engages either a different or more intensive quality control mechanism than the E200K mutant. Because the protein levels are reduced much more than the mRNA levels, the protein misfolding may be happening during and after mRNA translation and both get triaged for degradation.

## SST[+] neurons show pronounced translatome changes in pre-symptomatic stages of CJD and FFI

A general characterization of translatome profiles for disease-targeted cell types in both disease models was done by differential gene expression analysis with the DESeq2 R package [30] (Table S2). Because the mice were at a pre-symptomatic disease stage, we expected mild changes to gene expression and therefore defined differentially expressed genes (DEGs) to have a false discovery rate (FDR) ≤ 0.05 without a log fold change (LFC) cutoff (Figs 4A and B and S3A and B). Surprisingly, SST[+] neurons responded with the highest number of DEGs in both disease models (CJD: 153, FFI: 684), whereas PV[+] neurons showed very few DEGs (CJD: 2, FFI: 3). A comparison of shared DEGs between cell types of the same disease revealed that most DEGs were unique to a given cell type, including GABAergic subtypes (Fig 4C and D). In contrast, SST[+] neurons demonstrated a high overlap of DEGs in CJD and FFI, with 55 down- and 58 up-regulated genes shared (Fig 4E). There were few shared genes in other cell types, likely affected by the overall low number of DEGs (Fig S3C). Because little is known about the vulnerability of SST[+] neurons to PrDs, many of our analyses focused on these interesting cells.

In both mutants, SST[+] neurons displayed increased expression of many ribosomal protein mRNAs: of 79 ribosomal proteins, 26 were up-regulated in CJD (mean log$_2$FC = 0.42, SD = 0.09) and 57 in FFI (mean log$_2$FC = 0.44, SD = 0.09) (Fig S3D). Besides suggesting an increased need to synthesize proteins, the high functional connectivity of these genes is strongly indicative of a coordinated response. To measure the coordination amongst other DEGs we performed overrepresentation analysis (ORA) looking for enriched

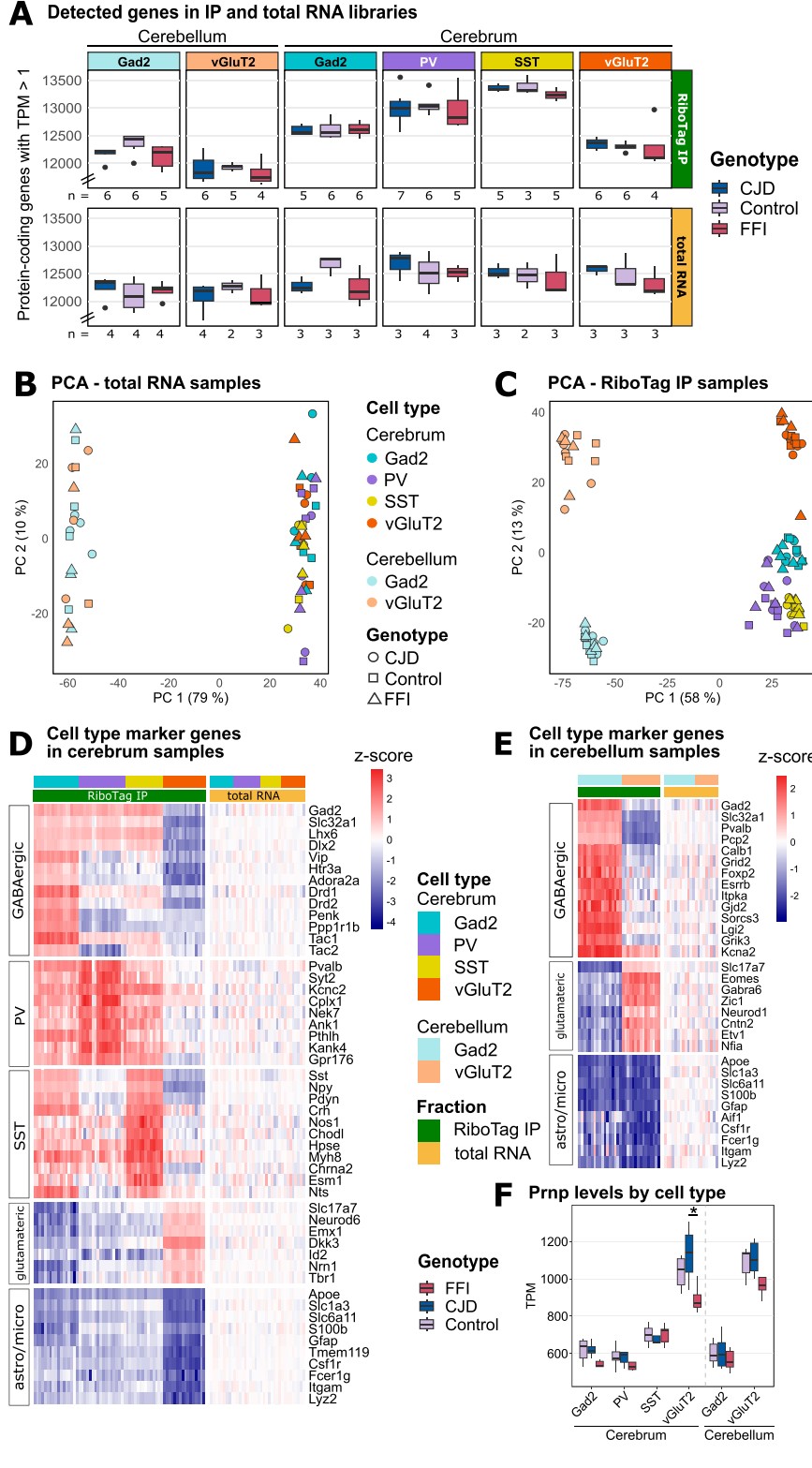

**Figure 3. Cell type–specific translatome isolation with RiboTag Immunoprecipitation (IP).**
**(A)** Number of expressed protein-coding genes in RiboTag-IP (upper panel) and total RNA samples (lower panel). The number of biological replicates for RiboTag IP samples is indicated below each box plot. Total RNA samples were prepared and sequenced for a subset of the mice used for RiboTag IPs. Note that the y-axis is broken to aid visualization. No significant differences between genotypes were detected. **(B)** Principal component analysis (PCA) of total RNA samples shows clustering by region but not cell type or genotype. **(C)** IP samples show clustering by targeted cell type but not genotype. PCA plots are based on top 1% most variable protein-coding genes. **(D, E)** Heat maps showing enrichment of cell type–specific marker genes in IP and total RNA samples obtained from cerebrum (D) and cerebellum (E). Row-wise Z scores were calculated based on transcripts per million (TPM) values across samples and normalized to input (total RNA) levels. **(F)** *Prnp* expression levels were comparable between cell types for mutant *Prnp* and control mice in GABAergic neurons (Gad2). Glutamatergic (vGluT2) neurons showed significantly lower *Prnp* expression levels in fatal familial insomnia mice compared with Creutzfeldt–Jakob disease (Kruskal–Wallis, $P$-value = 0.026, $\chi^2$ = 7.312); pairwise post hoc test: *, Dunn test, FDR < 0.05. Note that the y-axis starts at 400 to aid visualization. Astro, astrocyte; FDR, false discovery rate; Micro, microglia; PCA, principal component analysis.

gene ontology (GO) terms using Fisher's exact test. Up-regulated DEGs In CJD SST[+] neurons were associated with translation (ribosomal protein genes), actin cytoskeleton, actin-filament organization, and axonogenesis (FDR ≤ 0.01, Fig S4 and Table S3). In FFI SST[+]

neurons, up-regulated DEGs were mostly related to translation (*Snu13*, *Eef1a1*, *Eef12*, and 56 ribosomal proteins) (Fig S5A and Table S3). Cytoskeleton and cell adhesion-related terms were enriched among both up- and down-regulated DEGs in FFI SST[+] neurons (up:

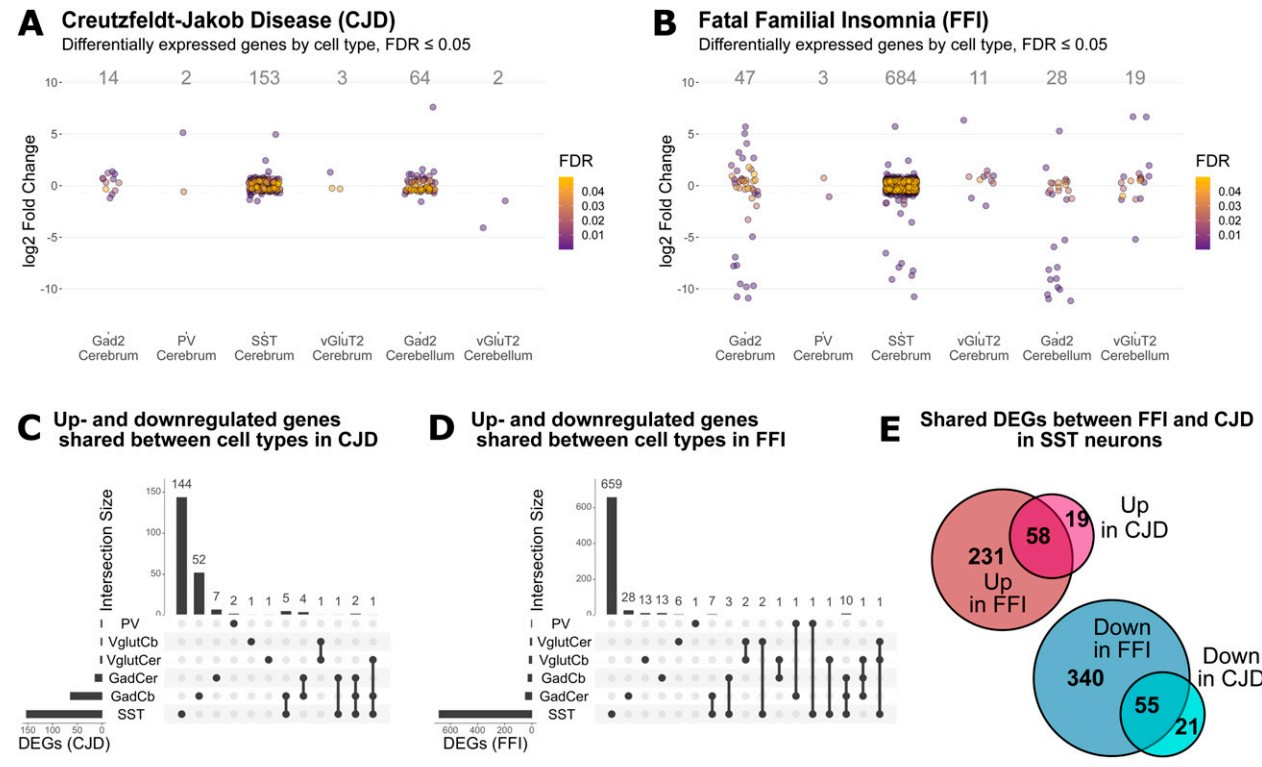

**Figure 4. Differential gene expression analysis reveals strong expression changes in SST⁺ neurons in Creutzfeldt–Jakob disease and fatal familial insomnia (FFI).**
**(A, B)** Zoom in of dotplots showing significantly (FDR ≤ 0.05) differentially expressed genes (DEGs) by cell type in Creutzfeldt–Jakob disease (A) and FFI (B), respectively. See Fig S3 for full-sized plots. **(C, D)** Number of DEGs shared between cell types. **(E)** Number of up- and down-regulated DEGs by disease models in somatostatin-expressing (SST⁺) neurons. FDR, false discovery rate.

"myelination," "actin-binding," "focal adhesion" and "cell-substrate junction"; down: "processes related to neurite morphogenesis and organization," "microtubule binding" and "motor activity," synaptic plasticity and ion-channels or receptor components) (Fig S5B). GTPase activity–related genes, such as activators of Rho-family GTPases (*Arhgap32,35,44*), Rho guanine nucleotide exchange factors (GEFs) (*Als2, Agap2, Trio,* and *Dock4*), and downstream effectors (*Cdc42bpa* and *Rock2*) were also overrepresented among downregulated DEGs in FFI SST⁺ neurons. Rho GTPases are known regulators of actin cytoskeleton dynamics (reviewed here: (31)), including dendritic spine formation and density (32), further indicating a high connectivity between DEGs. Collectively, these results suggest a concerted effort to reorganize the cytoskeleton of SST⁺ neurons. In summary, CJD and FFI showed a surprisingly high overlap in DEGs and, to a lesser extent, in enriched GO terms, suggesting that these neurons activate similar responses in both diseases.

### Gene set enrichment analysis (GSEA) characterizes specific responses by each cell type

A limitation of ORA is that coordinated expression changes of several genes within a pathway may be biologically important but would be excluded if the individual changes were statistically insignificant. Therefore, we applied a complementary approach, GSEA (33), to assess enrichment of GO terms for biological processes (BP) and KEGG pathways in each cell type, using the piano R package (34) which provides consensus enrichment scores, summarizing results

of six statistical methods. In addition, separate *P*-values for different directionalities of change were provided for each gene set (Table S4). Besides finding additional pathways, analyzing the data this way can identify cells responding similarly to both diseases as those having an abundance of terms that are changed in the same direction in both diseases, and can identify cells responding differently between the diseases as those that have an abundance of terms that are changed in both diseases but in opposite directions or are changed in only one disease (see Fig 5 for condensed results, Fig S6 for complete results).

Upon completing the GSEA, we first examined the results of SST⁺ neurons, the cells with the most DEGs, and found the top ranked gene sets involved up-regulation of translation-related pathways and ND-related pathways. Shared down-regulated terms included "neuron differentiation" and neurite-related terms ("axon extension," "positive regulation of neuron projection development," and "synapse organization"). FFI SST⁺ neurons also showed downregulation of pathways and terms related to synaptic function, "phosphatidylinositol phosphorylation," and "small GTPase mediated signaling transduction" (Fig 5A, column 3). This analysis indicated a broadly similar response of SST⁺ neurons in both disease models, with 15 terms changed with the same directionality, and no terms with opposite directionality (Fig 5B). Therefore, the results of this GSEA reflect DESeq and ORA results for SST⁺ neurons, indicating this is a robust method for these samples.

Interestingly, like SST⁺ neurons, PV⁺ neurons (Fig 5A, column 2) showed shared up-regulation of terms related to translation and

## A   Selected GO terms (Biological Process) and KEGG pathways

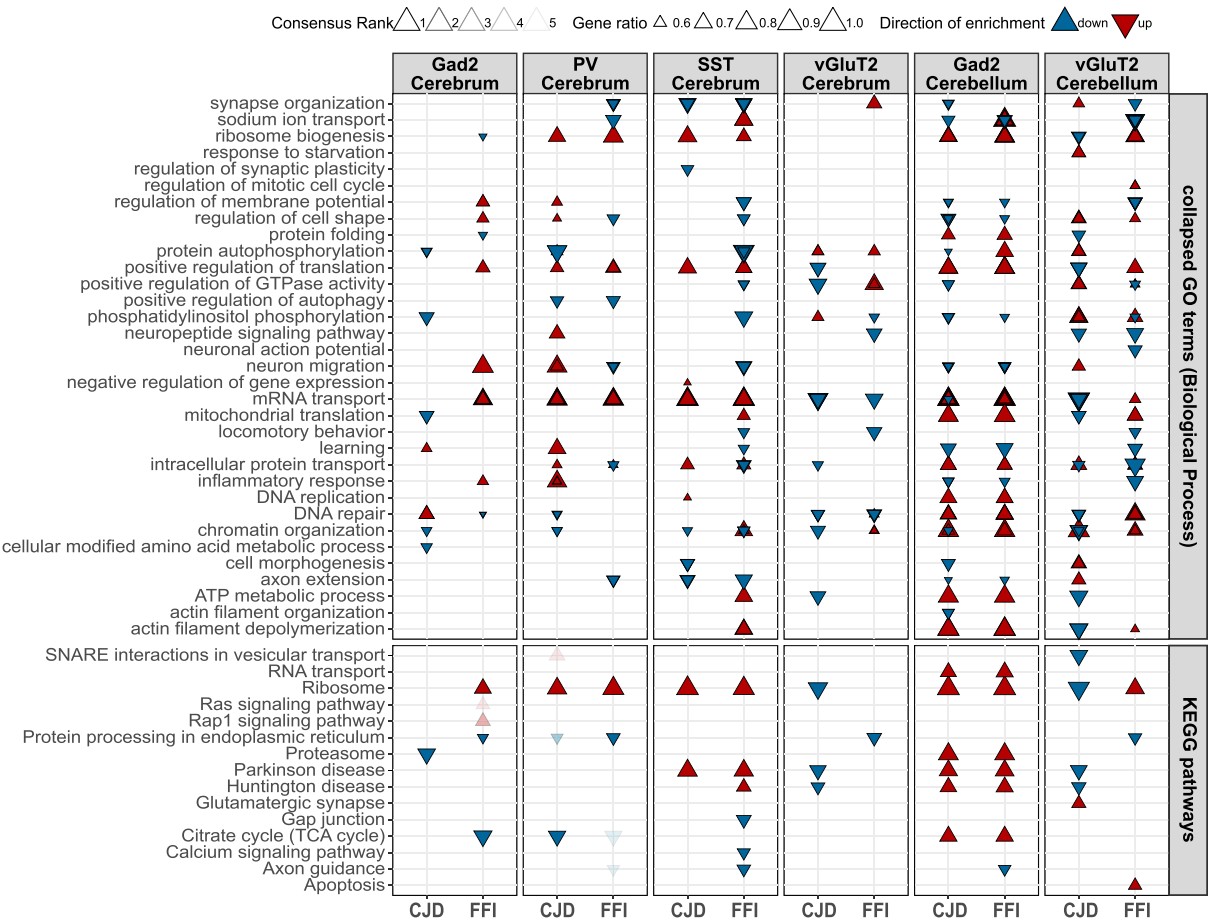

## B   Overview of shared and unique terms for CJD and FFI

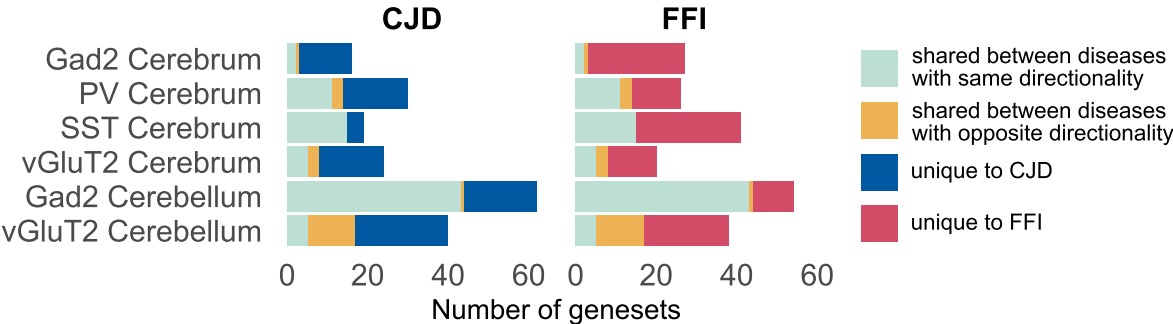

**Figure 5.   Gene set enrichment analysis (GSEA) reveals targeted GABAergic cell types have similar responses in Creutzfeldt–Jakob disease and fatal familial insomnia.**
**(A)** GSEA results for selected GO Biological Process terms and KEGG pathways. Extended plot in Fig S6. Consensus scores were calculated based on results from GSEA methods providing gene set statistics for distinct up- and down-regulated gene sets, represented by upward and downward arrowheads, respectively. Selected terms with FDR ≤0.05 in at least three of the six applied GSEA methods and a consensus rank of ≤5 are displayed. The gene ratio indicates the proportion of genes changed in the indicated direction relative to the total number of genes in the set. For visualization, GO terms were collapsed based on semantic similarity (method = "Resnik," threshold = 0.8) to reduce redundancy. This resulted in GO results in some cases displaying metrics for several gene sets summarized under the listed parent term. **(B)** Comparison of terms occurring in one or both disease models by cell type suggested cerebellar Gad2[+], SST[+], and PV[+] neurons show similar responses between both disease models, whereas cerebellar vGluT2[+] neurons show the most terms changed in one disease and either unchanged or changed in the opposite directions, suggesting a more disease-specific response. Plot shows the number of pathway or collapsed (parent) terms which were up- or down-regulated in only one disease and unchanged in the other (unique, blue or red), occurred in both diseases but with mixed or opposing directionalities (orange) or were shared between diseases showing the same directionality (light green).

down-regulation of "positive regulation of autophagy" and "protein processing in the endoplasmic reticulum" (ER). This insight was missed by DESeq and ORA because of the low number of DEGs (Fig 4A and B). Down-regulated GO terms exclusive to FFI PV+ neurons suggested a disruption in synaptic function ("synapse organization," "Axon guidance," and "positive regulation of neuron projection development"). Interestingly, gene sets "neuron migration," "neuron differentiation," and "regulation of cell shape" were up-regulated in CJD but down-regulated in FFI PV+ neurons (Fig 5A, column 2). In total there were 11 terms changed in the same direction in both diseases but there were also three terms that changed in opposite directions, and 12 or 16 terms that were specific to one disease, suggesting PV+ neurons had a mix of both similar and dissimilar responses (Fig 5A and B).

Whereas SST+ neurons had highly similar enrichment patterns between the diseases, and PV+ neurons had a mixed pattern, Gad2+ neurons of the cerebrum were dissimilar between the disease models, being the cells with the fewest terms shared in both models (n = 2) and having a large number of unique terms (CJD = 13; FFI = 24) (Fig 5B). In FFI Gad2+ neurons, terms related to ribosome pathway, GTPase signaling (Ras and Rap1), neuron migration and inflammation were up-regulated, whereas in CJD Gad2+ neurons terms related to metabolic processes, mitochondrial translation, proteasome, and DNA repair were down-regulated (Fig 5A, column 1). The final cell type from the cerebrum studied, the vGluT2+ neurons, showed a mixed response with three terms in opposite directions and five terms in the same direction. Cerebral vGluT2+ neurons also showed a CJD-specific down-regulation of ND-related pathways, as well as "ribosome" and "positive regulation of translation." This was notable as we observed that these translation-related terms were up-regulated in all cell types in FFI, with the exception of cerebral vGluT2+, where they were unaffected, and were also up-regulated in PV+, SST+, and cerebellar Gad2+ neurons in CJD. This suggests that excitatory neurons show a disease and cell type–specific response in their regulation of the translational machinery.

Like in the cerebrum, cell types in the cerebellum showed specific responses. Across all six cell types studied, Gad2+ neurons of the cerebellum exhibited the most terms with the same directionality (n = 43) and only 1 term with opposite directionality (Fig 5B), indicating a very similar response between diseases. In both disease models, Gad2+ cerebellar neurons showed up-regulation of terms related to translation, splicing, RNA and protein transport, and ND related pathways, whereas GO terms and pathways related to phosphatidylinositol and GTPase signaling, inflammation, and cellular morphology ("regulation of cell shape" and "Cell adhesion molecules"), neuron migration and differentiation were down-regulated in both diseases. In sharp contrast to the similar changes in cerebellar Gad2+ neurons, changes in cerebellar vGluT2+ neurons were disease-specific. Of the six cell types studied, vGluT2+ neurons had the most shared terms with opposite directionality (n = 12) and the second fewest terms with the same directionality (n = 5, Fig 5B). The most prominent terms with opposite directions were related to translation, mRNA transport, and DNA repair, which were down in CJD but up in FFI. Prominent terms unique to CJD vGluT2+ cerebellar neurons included down-regulation of ND pathways, splicing, protein folding, and starvation response. In contrast,

prominent terms unique to FFI vGluT2+ cerebellar neurons involved up-regulation of apoptosis and mitotic cell cycle, and down-regulation of ER protein processing and synaptic function (Fig 5A, column 6). Thus, in addition to finding interesting pathways, GSEA revealed that every cell type had a unique response and that in some cell types the two diseases caused different, disease-specific responses, whereas in other cell types, there were similar responses, and in still others there were mixed responses.

## Identification of functional modules in an SST+ co-expression network

Because SST neurons are understudied in PrD research, we wondered if they might reveal new insights into therapeutic targets. Thus, we used a network-based approach to further elucidate patterns in gene expression changes in SST+ neurons. Using our SST+ neuron-specific translatome data we constructed an undirected weighted gene co-expression network using pairwise gene correlations (FDR ≤ 0.01, Spearman $\rho$ > 0.82) (Table S5). Community analysis using the Leiden algorithm (35) generated six major modules (ranging in size from 249 to 2,733 genes) consisting of genes with highly correlated expression patterns across all conditions (Fig 6A), which were validated by comparison to a random network. As co-expression analysis builds on the assumption that correlation patterns between genes reflect functional connection, we used ORA to determine significantly enriched (FDR ≤ 0.01) ontology terms and pathways among module genes (Table S6).

Module 1 consisted predominantly of genes down-regulated in both diseases (Fig 6A), including 241 genes also differentially expressed, and predominantly down-regulated, in FFI. Module genes were significantly overrepresented (FDR ≤ 0.01) among terms related to synaptic transmission, protein modifications and transport, response to starvation, neuron projection development, and axon guidance. Module genes annotated to these terms also included several genes which we identified as differentially expressed either in both diseases (indicated in bold italics in Fig 6A) or specific for FFI (italics). Genes annotated to synapse organization, chromatin remodeling, and regulation of dephosphorylation-related terms included FFI-specific DEGs. Interestingly, ORA of module 1 genes also revealed autophagy-regulation ("negative regulation of macroautophagy" and "TORC1 signaling") and chromatin modifications ("positive regulation of histone ubiquitination") among the top enriched ontologies (Fig S7A).

Module 2 genes were enriched for translation, ribosomal biogenesis, and mitochondrial organization (Figs 6A and S7B). This is consistent with ORA results from up-regulated DEGs identified in CJD and FFI SST+ neurons (Figs S4 and S5), as module 2 contains ribosomal protein genes, a large percentage of which were up-regulated in both diseases. Additional enriched GO terms related to ER stress, regulation of apoptotic process-related terms, and unfolded protein response, which included several FFI-specific DEGs such as *activating transcription factors* 4 and *5*, *Atf4* and *Atf5* (Fig 6A). These results are consistent with those from GSEA (Fig 5) and functional analysis of cell type specific DEGs (Figs S4 and S5). This indicates that genes in modules 1 and 2 might be of particular interest to genetic PrD-associated pathological processes as they show highly correlated expression patterns with a high percentage

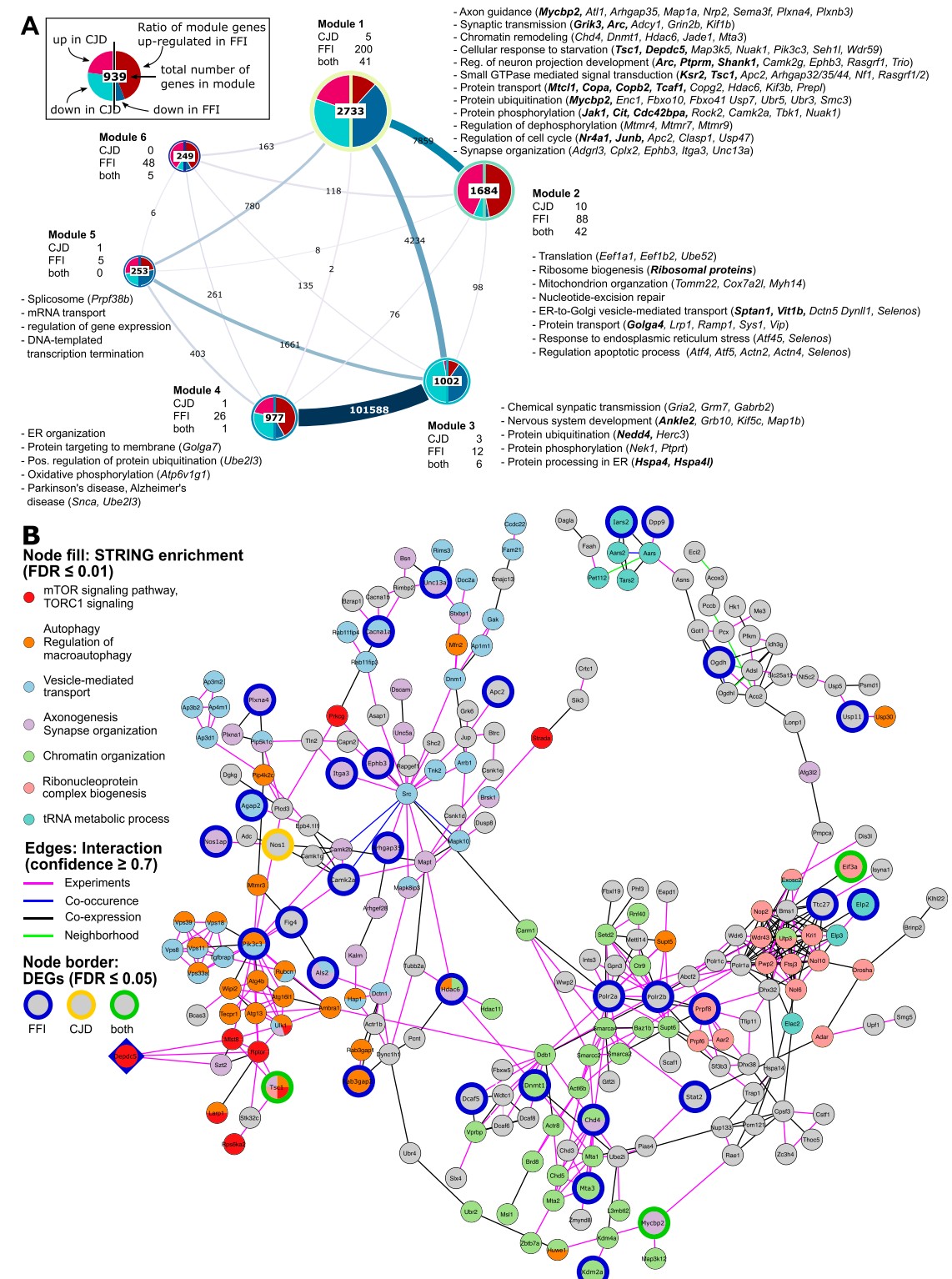

**Figure 6. Network analysis of SST⁺ neurons reveals mTOR signaling as a central regulator of expression changes.**
**(A)** Visualization of identified modules for a somatostatin (SST⁺) neuron weighted gene co-expression network with edges indicating inter-module connections. Half-pie charts displayed over the network nodes indicate the ratio of up- and down-regulated genes in Creutzfeldt–Jakob disease (left half) and fatal familial insomnia (FFI) (right half). Labels indicate the number of module genes differentially expressed in Creutzfeldt–Jakob disease, FFI or both diseases. Significantly enriched GO terms (FDR ≤ 0.01) were collapsed by semantic similarity with selected parent terms, or enriched KEGG pathways (FDR ≤ 0.01) displayed in module annotations. Gene symbols in parentheses show a selection of annotated genes significantly differentially expressed in FFI (italics) or both diseases (bold italics). **(B)** The largest connected component

of DEGs and are functionally closely related to identified dysregulated terms.

Genes in Module 3 were mostly down-regulated in both diseases and functionally associated with chemical synaptic transmission, nervous system development, and protein modifications (Fig 6A) but also included translation initiation, regulation of macroautophagy, and stress granule assembly among top enriched GO terms (Fig S7C). Module 4 was highly connected with Module 3 and contained predominantly up-regulated genes associated with ER organization, protein targeting and ubiquitination (Figs 6A and S7D). Module 4 also contained several mitochondrial genes associated with KEGG pathways oxidative phosphorylation, thermogenesis and Alzheimer's and Parkinson's disease pathways (Fig 6A). Module 5 genes showed significant overrepresentation of terms related to mRNA splicing and RNA processing (Figs 6A and S7E). No significant enrichment was detected for genes in module 6. This co-expression network analysis further supports the notion that changes in SST⁺ neurons were highly coordinated and remarkably similar in FFI and CJD brains.

### Hub genes point towards two potential therapeutic targets

To find potentially important regulators, we next identified hub genes that display the largest number of co-expressed genes. We defined hubs as the top 1% of genes with the highest degree of centrality, that is, most direct neighbors, in each module of our co-expression network (Table 1). Notably, three hub genes in Module 1 were also differentially expressed in FFI: GATOR1 subunit *Depdc5* (*DEP Domain containing Complex 5*; degree: 560), histone-deacetylase *Mta3* (Metastasis Associated 1 Family Member 3; degree: 551) a subunit of the nucleosome remodeling and deacetylase (NuRD) complex, and *Gtf3c1* (General Transcription Factor IIIC Subunit 1; degree: 585) a mediator of RNA polymerase III transcription. Because down-regulation of these highly connected hub genes suggests they have a central role in the pathological process that may have far-reaching effects on interaction partners, we next aimed to further validate the interaction of hub genes with their co-regulated neighbors. For this we constructed a protein–protein interaction (PPI) network for each hub gene and its first-degree neighbors, to determine whether known interactions between products of co-regulated genes exist. Predicted PPIs were obtained from STRINGdb, considering only interactions with a combined confidence score ≥0.7, and excluding interactions based on text mining and databases.

There were no predicted interactions of *Gtf3c1* with its co-regulated direct neighbors, indicating that this method did not provide further insight for this gene. However, the PPI network for *Mta3* included 220 of 551 co-regulated genes from our topological network (Fig S8), whereas the PPI network for *Depdc5* included 230 of 560 co-regulated genes (Figs 6B and S9). Both networks additionally showed strong overlap with 145 shared genes and included 30 genes significantly down-regulated in FFI (Fig 6B, blue border) or two in both diseases (green border). Pathway and GO enrichment analysis using the STRING Enrichment application (FDR ≤ 0.05)

revealed association of *Depdc5* PPI-network genes with autophagy, chromatin organization, vesicle-mediated transport, and neurite morphology (axonogenesis and synapse organization), ribonucleoprotein complex biogenesis, and tRNA metabolic process. *Depdc5* and its direct neighbors in the PPI network were associated with TORC1 signaling. Given the far-reaching effects of mTOR signaling on metabolic regulation and autophagy, its involvement in ageing and proposed involvement in neurodegeneration, we propose this may be a central regulator behind translatome changes we observed in SST⁺ neurons in genetic PrDs. Taken together, this analysis indicates that for both diseases SST⁺ neurons show the largest response with TORC1 signaling posing a potential underlying regulatory mechanism.

## Discussion

Here we report cell type–specific responses in knock-in mouse models of two genetic PrDs at a pre-symptomatic stage, by translatomic analysis of vGluT2, Gad2, PV, and SST expressing neurons in the cerebrum, and vGluT2 and Gad2 expressing neurons of the cerebellum. Because the thalamus and hippocampus are the brain regions most affected in FFI and CJD, are two of the regions with the highest *Prnp* expression (29), and are almost exclusively glutamatergic, we expected the vGluT2 neurons in the cerebrum to have the highest number of DEGs. We were therefore surprised that vGluT2 samples from the cerebrum, even though they expressed the highest levels of *Prnp* (Fig 3F), had very few DEGs. We also expected each cell type to have a unique response to each disease and were again surprised that some of the cell types had similar responses to both diseases. For example, in both disease models SST⁺ neurons had the highest number of DEGs, and 74% of CJD DEGs were shared with FFI. Furthermore, additional levels of similarity between CJD and FFI emerged from ORA and GSEA studies, for example, up-regulation of translation, ND pathways, and actin-binding proteins, suggesting the disease mechanisms are similar in both models. Because SST⁺ neurons showed more DEGs in FFI than in CJD, and a functional analysis revealed down-regulation of genes related to synaptic function and GTPase signaling, SST⁺ neurons appear to be at a more advanced disease stage in FFI than CJD brains.

Previous reports indicated a pronounced early loss of cortical PV⁺ neurons in patients and models of sporadic, genetic and variant CJD, although they are relatively spared in FFI patients (36). Based on their reported early vulnerability and the hypothesis that transcriptional changes precede neuronal pathology and loss (18), we expected to observe more pronounced gene expression changes in PV⁺ neurons, at least for CJD mice, but this was not apparent based on the number of DEGs. However, GSEA of FFI PV⁺ neurons revealed enrichment in gene sets that could be associated with neuronal dysfunction, such as ER protein processing or synapse organization, suggesting PV⁺ neurons are mildly affected early in this model. Importantly, we observed similar, although more

---

of a protein–protein interaction network for module 1 hub gene *Depdc5* (diamond-shaped) and first-degree neighbors. Larger nodes with colored borders indicate DEGs by disease. Node colors show selected functional associations of genes based on STRINGdb enrichment (FDR ≤ 0.01). DEG, differentially expressed gene; FDR, false discovery rate; SST, somatostatin.

**Table 1:** Hub genes for SST neuron co-expression network.

| Gene | Module | DEG | Description | Entrez ID | LFC (FFI) | FDR (FFI) |
|---|---|---|---|---|---|---|
| Depdc5 | 1 | FFI | DEP domain containing 5 | 277854 | −0.4078 | 0.0096 |
| Gtf3c1 | 1 | FFI | General transcription factor III C 1 | 233863 | −0.3991 | 0.0189 |
| Mta3 | 1 | FFI | Metastasis associated 3 | 116871 | −0.3837 | 0.0215 |
| Abcf2 | 1 | Neither | ATP-binding cassette, sub-family F (GCN20), member 2 | 27407 | −0.191 | 0.2543 |
| Abcg4 | 1 | Neither | ATP binding cassette subfamily G member 4 | 192663 | −0.1572 | 0.3968 |
| Abr | 1 | Neither | Active BCR-related gene | 109934 | −0.1583 | 0.3333 |
| Adar | 1 | Neither | Adenosine deaminase, RNA-specific | 56417 | −0.2753 | 0.0665 |
| Cabin1 | 1 | Neither | Calcineurin binding protein 1 | 104248 | −0.2813 | 0.1105 |
| Carm1 | 1 | Neither | Coactivator-associated arginine methyltransferase 1 | 59035 | −0.1128 | 0.5449 |
| Dab2ip | 1 | Neither | Disabled two interacting protein | 69601 | −0.1931 | 0.2543 |
| Dusp8 | 1 | Neither | Dual specificity phosphatase 8 | 18218 | −0.1254 | 0.514 |
| Gba2 | 1 | Neither | Glucosidase beta 2 | 230101 | −0.2639 | 0.1263 |
| Kcnq2 | 1 | Neither | Potassium voltage-gated channel, subfamily Q, member 2 | 16536 | −0.2268 | 0.1584 |
| Kdm4a | 1 | Neither | Lysine (K)-specific demethylase 4A | 230674 | −0.3135 | 0.0563 |
| Kif1a | 1 | Neither | Kinesin family member 1A | 16560 | −0.3211 | 0.0624 |
| Map3k4 | 1 | Neither | Mitogen-activated protein kinase kinase kinase 4 | 26407 | −0.2906 | 0.0816 |
| Nol6 | 1 | Neither | Nucleolar protein family 6 (RNA-associated) | 230082 | −0.2643 | 0.1356 |
| Osbp2 | 1 | Neither | Oxysterol binding protein 2 | 74309 | −0.2788 | 0.089 |
| Pdxk | 1 | Neither | Pyridoxal (pyridoxine, vitamin B6) kinase | 216134 | −0.2316 | 0.1489 |
| Pdzd4 | 1 | Neither | PDZ domain containing 4 | 245469 | −0.2219 | 0.1808 |
| Ptprs | 1 | Neither | Protein tyrosine phosphatase, receptor type, S | 19280 | −0.267 | 0.1095 |
| Rap1gap | 1 | Neither | Rap1 GTPase-activating protein | 110351 | −0.2164 | 0.2388 |
| Rusc2 | 1 | Neither | RUN and SH3 domain containing 2 | 100213 | −0.331 | 0.054 |
| Smarca2 | 1 | Neither | SWI/SNF related, matrix associated, actin dependent regulator of chromatin, subfamily a, member 2 | 67155 | −0.1895 | 0.2628 |
| Tecpr1 | 1 | Neither | Tectonin beta-propeller repeat containing 1 | 70381 | −0.2488 | 0.1649 |
| Vps11 | 1 | Neither | VPS11, CORVET/HOPS core subunit | 71732 | −0.2809 | 0.0974 |
| Wdr81 | 1 | Neither | WD repeat domain 81 | 192652 | −0.24 | 0.1874 |
| Bola2 | 2 | Neither | bolA-like 2 (Escherichia coli) | 66162 | 0.1598 | 0.3747 |
| Cbarp | 2 | Neither | Calcium channel, voltage-dependent, beta subunit associated regulatory protein | 100503659 | 0.1118 | 0.5978 |
| Chgb | 2 | Neither | Chromogranin B | 12653 | −0.0514 | 0.8321 |
| Clstn1 | 2 | Neither | Calsyntenin 1 | 65945 | −0.0957 | 0.6549 |
| Clstn3 | 2 | Neither | Calsyntenin 3 | 232370 | 0.0567 | 0.8178 |
| Eno2 | 2 | Neither | Enolase 2, gamma neuronal | 13807 | 0.0224 | 0.9305 |
| Erp29 | 2 | Neither | Endoplasmic reticulum protein 29 | 67397 | 0.2004 | 0.2832 |
| Mlf2 | 2 | Neither | Myeloid leukemia factor 2 | 30853 | 0.1074 | 0.582 |
| Mtch1 | 2 | Neither | Mitochondrial carrier 1 | 56462 | 0.1217 | 0.5536 |
| Ndufaf2 | 2 | Neither | NADH:ubiquinone oxidoreductase complex assembly factor 2 | 75597 | 0.1683 | 0.3876 |
| Nomo1 | 2 | Neither | Nodal modulator 1 | 211548 | −0.0515 | 0.8354 |
| Pomgnt2 | 2 | Neither | Protein O-linked mannose beta 1,4-N-acetylglucosaminyltransferase 2 | 215494 | −0.0482 | 0.8468 |

**Table 1: Continued**

| Gene | Module | DEG | Description | Entrez ID | LFC (FFI) | FDR (FFI) |
|------|--------|-----|-------------|-----------|-----------|-----------|
| Psmd4 | 2 | Neither | Proteasome (prosome, macropain) 26S subunit, non-ATPase, 4 | 19185 | 0.1939 | 0.2965 |
| Rab3a | 2 | Neither | RAB3A, member RAS oncogene family | 19339 | 0.1032 | 0.6085 |
| Tmsb10 | 2 | Neither | Thymosin, β 10 | 19240 | 0.1508 | 0.4312 |
| Tomm7 | 2 | Neither | Translocase of outer mitochondrial membrane 7 | 66169 | 0.1699 | 0.3821 |
| Abca5 | 3 | Neither | ATP-binding cassette, sub-family A (ABC1), member 5 | 217265 | −0.0966 | 0.6559 |
| Appbp2 | 3 | Neither | Amyloid beta precursor protein (cytoplasmic tail) binding protein 2 | 66884 | −0.0543 | 0.8261 |
| Cdc27 | 3 | Neither | Cell division cycle 27 | 217232 | −0.0475 | 0.8426 |
| Dpp10 | 3 | Neither | Dipeptidylpeptidase 10 | 269109 | −0.0441 | 0.8597 |
| Dzip3 | 3 | Neither | DAZ interacting protein 3, zinc finger | 224170 | −0.0347 | 0.8885 |
| Mctp1 | 3 | Neither | Multiple C2 domains, transmembrane 1 | 78771 | −0.0424 | 0.8659 |
| Nampt | 3 | Neither | Nicotinamide phosphoribosyltransferase | 59027 | −0.0451 | 0.854 |
| Nr1d2 | 3 | Neither | Nuclear receptor subfamily 1, group D, member 2 | 353187 | −0.0234 | 0.9282 |
| Rab1a | 3 | Neither | RAB1A, member RAS oncogene family | 19324 | 0.0018 | 0.9937 |
| Zdhhc17 | 3 | Neither | Zinc finger, DHHC domain containing 17 | 320150 | −0.0592 | 0.8072 |
| Azin1 | 4 | Neither | Antizyme inhibitor 1 | 54375 | −0.0302 | 0.9057 |
| Negr1 | 4 | Neither | Neuronal growth regulator 1 | 320840 | −0.0148 | 0.9557 |
| Pten | 4 | Neither | Phosphatase and tensin homolog | 19211 | −0.034 | 0.8881 |
| Rab10 | 4 | Neither | RAB10, member RAS oncogene family | 19325 | −0.0315 | 0.9035 |
| Rab14 | 4 | Neither | RAB14, member RAS oncogene family | 68365 | −0.0073 | 0.98 |
| Septin7 | 4 | Neither | Septin 7 | 235072 | −0.0231 | 0.9258 |
| Slc25a16 | 4 | Neither | Solute carrier family 25 (mitochondrial carrier, Graves disease autoantigen), member 16 | 73132 | 0.0129 | 0.9613 |
| Slc38a2 | 4 | Neither | Solute carrier family 38, member 2 | 67760 | −0.0063 | 0.9823 |
| Slc6a15 | 4 | Neither | Solute carrier family 6 (neurotransmitter transporter), member 15 | 103098 | 0.0247 | 0.9254 |
| Ccdc82 | 5 | Neither | Coiled-coil domain containing 82 | 66396 | −0.2945 | 0.096 |
| Ythdc1 | 5 | Neither | YTH domain containing 1 | 231386 | 0.0531 | 0.8308 |
| Hpcal4 | 6 | Neither | Hippocalcin-like 4 | 170638 | 0.0515 | 0.7933 |
| Parva | 6 | Neither | Parvin, alpha | 57342 | −0.0184 | 0.9347 |

Hub genes were defined as top 1% of genes with highest degree centrality in each module. The last columns indicate $\log_2$ fold changes of hub genes and adjusted *P*-value (FDR) in FFI samples compared with wild-type controls.

intense, changes in SST+ neurons, a cell type that has previously not been implicated in PrD pathology.

In contrast, GSEA results for cerebellar neurons displayed widespread changes of major pathways and functional processes, despite few DEGs detected for vGluT2+ neurons in CJD and both neuron types in FFI. This suggests there is a moderate but coordinated response, in line with early neuropathological changes in the cerebellum observed in both diseases. Our analyses showed high similarities in enriched terms (and their directionalities) between disease models in cerebellar Gad2+ neurons, suggesting shared mechanisms underlying the pathology in these cells. In contrast, cerebellar vGluT2+ neurons showed disease-specific responses. Further studies to confirm these results and determine their role for cerebellar pathology would be well placed. Overall, our findings indicate that SST+ neurons are a previously unrecognized neuronal subtype affected early in FFI and CJD. Because vulnerability of SST+ subpopulations has been described in other NDs ([11]), but not in PrDs, a deeper exploration of how these neurons responded in FFI and CJD was performed.

## mTORC1 inhibitors are down-regulated in SST+ neurons

To further elucidate potential mechanisms and regulatory factors underlying the observed translatome changes in SST+ neurons, we sought to identify topological modules and central genes by constructing a weighted gene-correlation network. These results indicated down-regulation of TORC1 inhibitors, which likely lead to SST+ neuron-specific activation of mTOR signaling, a positive regulator of protein synthesis, synaptogenesis, and negative regulator

of autophagy. Thus, this single pathway may be responsible for many of the DEGs in SST⁺ neurons.

Genes central in regulating TORC1 activity were differentially expressed in FFI SST⁺ neurons, and the main TORC1 inhibitor, *Tsc1*, was down-regulated in both disease models. Moreover, SST⁺ neurons in both models showed expression changes consistent with increased mTOR activity, including up-regulated expression of ribosomal and mitochondrial genes, down-regulation of autophagy, and cytoskeletal reorganization. Topological network analysis of SST⁺ samples indicated *Depdc5* as one of the module 1 hub genes. *Depdc5*, which was also significantly down-regulated in FFI, encodes a subunit of the TORC1 inhibitor complex GATOR1, involved in amino acid–dependent TORC1 activation, and is associated with epilepsy. Haploinsufficiency of *Depdc5* in induced pluripotent stem cells causes aberrant morphology and TORC1 hyperactivation ([37]), suggesting that down-regulation of *Depdc5* in our model may have considerable impact, despite the small fold change. Recent publications revealed that loss of Tsc1–mediated mTOR inhibition results in a shift of electrophysiological properties in a subset of SST⁺ cortical interneurons ([38]), suggesting mTOR activity in maintenance of cell identity in SST⁺ neurons. Histone deacetylase 6, *Hdac6*, which was down-regulated in FFI and strongly co-expressed with *Depdc5*, is suggested to be a modulator of TORC1 signaling ([39]) and central in inducing autophagy as a compensatory mechanism for impaired ubiquitin-proteasome system degradation ([40]). Overexpression of *Hdac6* in cortical neurons exposed to a toxic PrP fragment (PrP¹⁰⁶⁻¹²⁶) was shown to increase cell survival by inducing autophagy through mTOR signaling modulation ([39]). Together, these points support the notion that mTOR signaling is affected in PrDs, and that this mechanism may be specific for SST neurons early in disease. Although it is conceivable that aberrant mTOR activity may affect maintenance of SST⁺ neuron identity and properties in the investigated diseases, additional research is required to investigate this and to determine how this would impact disease phenotype.

Aberrant activity of mTOR signaling has been demonstrated in many NDs, including Alzheimer's disease, Parkinson's disease, Huntington's disease, and PrDs ([41], [42]), and the central role of mTOR activity together with the availability of approved mTOR inhibitors, such as rapamycin and derivatives, has made it an attractive drug target in the search for a treatment for neurodegeneration. Although studies report positive effects of mTOR inhibitor treatment on cognition by enhancing autophagy and promoting clearance of protein aggregates, its overall role in neurodegeneration is more complex. mTOR activity is an example of antagonistic pleiotropy—showing beneficial effects early in life at the expense of negative effects later in life——by promoting synaptogenesis during youth at the expense of increased risk of damage by protein accumulation due to autophagy inhibition (reviewed in reference [43]). This, together with reported cell type–specific effects of mTOR activation ([44]) complicates using mTOR inhibition as a therapeutic strategy as its beneficial effects will likely depend on correct timing in the disease progression ([45]). Our data further highlight that cell type–specific differences in mTOR activity play an important role on whether intervention of mTOR inhibition results in overall positive or negative effects and therapies may need to be targeted to specific cell types.

## Limitations of this study

There are several limitations of our study that warrant consideration. First, the mice were backcrossed onto the 129S4 background which could result in linked genetic modifiers impacting our results. Indeed, the signal regulatory protein α (Sirpa) is only 2 million base pairs away from *Prnp* and was reported to drive a phenotype in *Prnp* knock-out mice ([46]) that was previously attributed to *Prnp*. This is of little concern for the EEG and sleep studies because the control and mutant mice carried knock-in alleles engineered the same way, but the RiboTag study used control mice carrying an unmodified wild-type allele. Although this is difficult to dismiss in our experiments, we think it is not a problem because both the strain of embryonic stem cells used to make the *Prnp* knock-in mouse lines, and the mouse strain they were backcrossed to, are 129 substrains. Indeed, Sirpa sequences in the mutant and control mice are identical. Furthermore, the DEGs were distributed randomly across the genome and were not enriched for being located within 40 million base pairs of *Prnp*, the maximum amount of flanking genome we estimate could remain in the knock-in mice.

A second limitation is that the expression changes were not validated with additional methods. Confirmation with histological labeling was not attempted because the relative fold changes were small and the signal from surrounding cells that express the gene of interest at stable levels would obscure the analysis. Nonetheless, there are multiple reasons to be confident in the overall results. First, we have a proper replicate size and the data were analyzed with multiple, distinct bioinformatics tools giving the same overall result. Second, a study of Huntington's disease using bacTRAP (a method analogous to RiboTag) showed the method is reliable because they found similar results with single cell analysis ([47]). Third, we have used this method to study Huntington's disease in parallel with the current study and acquired prion disease samples in a separate experiment ([18]), and both times found different cell types changing unique pathways, indicating that the results are disease model-dependent, as expected.

A third limitation is that we studied changes to the translatome but it is unknown if these were due to changes at the translation or transcription level because it is not possible to obtain information on total mRNAs from the specific cell types with RiboTag. Future studies could address this limitation by using Tagger mice ([48]). In addition to tagging ribosomes, Tagger also enables the capture of microRNAs, nuclei, and total RNAs. The last two components could be used to examine changes at the transcription level.

Finally, we intended to address the topic of selective vulnerability, but practical matters constrained us to study a limited number of brain regions and cell types. One constraint was that the regions selectively targeted in FFI and CJD mice, the thalamus and hippocampus, respectively, are difficult to dissect precisely (especially the thalamus) and were therefore studied together as components of the cerebrum samples. This complex mixture of regions likely obscured differences that exist. Nonetheless, with this approach we determined that SST neurons are highly changed, and because they are scarce in the thalamus and hippocampus, we would have missed this discovery had we focused on those regions. The number of cells to study was constrained because the intercross to obtain experimental mice required mice that were double homozygous for *Prnp* and either Cre or RiboTag genes, requiring several rounds of breeding and lots of mice and mouse room space.

Therefore, we focused on a limited set of neurons and changes to non-neuronal cells remain unknown. Considering the results reported here for neurons of FFI and CJD mice, and from a separate study of an acquired prion disease model studying neurons and astrocytes (18), future studies of FFI and CJD mice examining more cell types, including astrocytes, microglia, oligodendrocytes, and neurovascular cells would likely provide a deeper understanding and bring additional value to the current data. However, in this case Tagger mice (48) could be used in place of RiboTag.

## Conclusion

Our results demonstrate a pronounced response of SST[+] neurons to early, pre-symptomatic stages of FFI, with a marked, coordinated up-regulation of mitochondrial and ribosome biogenesis-associated genes and down-regulation of cytoskeletal proteins or regulator genes. We identified 67 candidate module hub genes in a co-expression network of SST[+] neurons, of which three (*Gtf3c1*, *Depdc5*, and *Mta3*) also showed differential expression in FFI and were therefore further validated. With a clear connection to mTOR signaling, a tentative pharmacologically targetable pathway can now be proposed for FFI and CJD and tested experimentally in the future.

We also report FFI and CJD to be more similar at the molecular level than predicted from differences in clinical signs and neuropathological changes. This was particularly true for SST[+] neurons, which have thus far been largely ignored in PrD research. Interestingly, we recently reported that SST[+] neurons showed little, if any, response to a widely studied mouse model (RML) of acquired PrD. Moreover, the genes and pathways changed in Gad2 and vGluT2 cells in RML-infected brains (18) were completely different from those in FFI and CJD brains, indicating that the genetic and acquired diseases are unexpectedly different. The largest differences between FFI and CJD were in vGluT2 neurons of the cerebellum, whereas vGluT2 cells of the cerebrum showed a mixed response. This mixed response may be related to the selective vulnerability reported previously because each disease causes neuropathological changes in different brain regions, especially the thalamus and hippocampus, both enriched in glutamatergic (vGluT2[+]) neurons. Because nearly all glutamatergic neurons in the cerebellum are granule cells, a very homogeneous cell type, changes there will not be obscured by non-responding cells. Because the cerebrum was a mixture of many regions, some affected and many not, the mixed response we observed for vGluT2[+] cerebrum samples would be expected. Similarly, GSEA of PV[+] neurons demonstrated a mix of terms, potentially reflecting the difference in vulnerability reported previously (9). A unifying explanation of the causes of selective vulnerability remains elusive but continued experimentation with methods like RiboTag (14) or Tagger (48) may help to eventually solve this mystery.

# Materials and Methods

## Mouse lines

129S4 mice homozygous for the mouse equivalent of the D178N (FFI) (15) or E200K (CJD) (16) substitution and the 3F4 epitope (L108M,

V111M) in the *Prnp* locus were studied. Because of deletion of a single codon at the N-terminus in the mouse *Prnp* gene, the mouse-equivalent nomenclature for these substitutions is D177N and E199K. All lines were backcrossed to the 129S4 background for at least eight generations. Cre and RiboTag mice were at least 99.8% 129S4 (details in reference 25), and the FFI, CJD and WT mice were back-crossed nine generations and therefore ~99.6% 129S4. This background was chosen because, unlike C57Bl/6 mice which can be hyperactive at night, 129S4 mice are relatively calm (25). We were concerned that mice with a non-uniform activity level would be prone to highly variable gene expression patterns.

For RNAseq experiments, mice homozygous for FFI (D178N-3F4) or CJD (E200K-3F4) or wild-type (unmodified) *Prnp* (Control) were crossed with RiboTag mice (B6N.129-Rpl22[tm1.1Psam]/J, line #011029; Jackson Laboratory) (14) to obtain mice double homozygous for *Prnp* and RiboTag. For cell type–specific targeting, double homozygotes where crossed with Cre-driver lines: Vglut2-IRES-Cre (23) (Slc17a6[tm2(cre)Lowl]/J, line #016963; Jackson Laboratory), Gad2-IRES-Cre (49) (Gad2[tm2(cre)Zjh]/J, line #010802; Jackson Laboratory), SST-IRES-Cre (49) (Sst[tm2.1(cre)Zjh]/J, line #013044; Jackson Laboratory), and PV-IRES-Cre (24) (B6;129P2-Pvalb[tm1(cre)Arbr]/J, line #008069; Jackson Laboratory). All experimental mice were therefore homozygous for mutant or unmodified *Prnp* and double heterozygous for Cre and RiboTag. Mice were euthanized at 9 mo of age (mean age 9.3 mo, SD: 0.7) by carbon dioxide asphyxiation between 10:00 AM and 14:00 PM, matching time points for PrD and control mice to minimize the influence of circadian rhythm–related gene expression changes. Brains were separated into hemispheres along the midline. From one hemisphere, the olfactory bulb was removed and discarded, the cerebellum was separated from the cerebrum, flash frozen in cryo-tubes on a dry-ice chilled metal block, and stored at –78°C. The second hemisphere was fixed in formalin.

## Electroencephalography (EEG) and sleep recordings

The EEG/Sleep studies reported here were contemporaneous with another recent study of acquired prion disease (18), and experimental details can be found in that report. EEG and electromyograph detection leads were implanted in the epidural layer of the frontal cortex or in the neck muscles, respectively, and were routed subcutaneously to connect to telemetric recorders (F20-EET; Data Sciences International) that were implanted in the intraperitoneal cavity. Sleep scoring and analysis was performed as reported before (50, 51, 52). 6-h sleep deprivations were accomplished with the gentle handling method. Although the mutant mice did not generally look unfit for surgeries, several mice assigned to the EEG study died or were euthanized just before, during or soon after the procedure, which may have biased the study. 1 WT and 1 FFI mouse were euthanized during surgery because of a large tumor/cyst in the abdomen. Other unplanned deaths for FFI mice included 1 euthanized because of inflammation of a leg 2 wk after surgery, four died during surgery, six died within a week after surgery, and two died later. Unplanned deaths for CJD mice included four that died during surgery, two died within a week after surgery, and one was killed before surgery because of poor health. Since no unexpected deaths occured in WT mice, some mutants may have an unrecognized condition that makes them less tolerant to surgeries. It is

possible these were more affected than the others and that the group successfully studied with EEG represents less affected mice. The final study group included 14 WT (9 females, mean age 20.6, SD 3.2 mo), 15 FFI (11 females, mean age 20.8, SD 1.6 mo), and 8 CJD (4 females, mean age 21.4, SD 1.2 mo), all carrying the 3F4 epitope.

## Buffers and preparation of tissue homogenates for RNA isolation

Stocks for polysome buffer (PSB; 50 mM Tris, pH = 7.5, 100 mM KCl, 12 mM MgCl$_2$, 1% IPEGAL CA-630 [Sigma-Aldrich], plus: 1 mM DTT, 60 U/ml RiboLock RNase inhibitor [Thermo Fisher Scientific], 100 µg/ml Cyclohexamide [Sigma-Aldrich], 2× SigmaFast EDTA-free protease inhibitor cocktail [Sigma-Aldrich] dissolved in PSB stock), high salt wash buffer (HSB; 50 mM Tris pH = 7.5, 300 mM KCl, 12 mM MgCl$_2$, 1% IPEGAL, plus: 1 mM DTT, 20 U/ml RiboLock, 100 µg/ml Cyclohexamide and 0.5× EDTA-free SigmaFast protease inhibitor cocktail), and extra high salt buffer (EHSB; HSB containing additional 300 mM NaCl) were prepared using RNase-free reagents and stored at 4°C. Inhibitors were added to stock solutions directly before use (indicated by "plus"). RNA purifications were performed in a Biosafety level 3 environment. Frozen tissue samples were homogenized at 450 rpm, using Wheaton Potter-Elvehjem homogenizers and PTFE pestles (DWK Life Science) with a motorized homogenizer (HEI-Torque Core, heidolph), in 200 µl ice-cold PSB per 0.01 g tissue. Homogenates were centrifuged at 4°C, 400$g$ for 2 min to collect nuclei. Supernatant was transferred to fresh vials and centrifuged at 4°C, 10,000$g$ for 10 min.

## Preparation of total RNA from tissue homogenates

After preparation of homogenates as described above, the supernatant (S1) was decanted and 200 µl (cerebrum) or 100 µl S1 (cerebellum) were used to purify total RNA by adding 300 µl Trizol (QIAGEN) and 300 µl chloroform, vigorous shaking, and incubation for 5 min at RT. Samples were centrifuged at 14,000$g$, RT, for 10 min; the aqueous phase was collected; mixed with 2× volume of 99.5% ethanol, 0.1× volume of 3 M sodium acetate (pH = 5.2), and GlycoBlue (Thermo Fisher Scientific; to 50 µg/ml final concentration); shaken vigorously; and incubated at –20°C for 2 h. Nucleic acids were pelleted by centrifugation at 18,000$g$, 4°C for 15 min. Supernatant was discarded and pellets incubated for 30 min in 350 µl PKB buffer (4 M guanidinium isothiocyanate, pH = 7.5, 0.1 M $\beta$-mercaptoethanol, 25 mM sodium citrate, and 0.5% Sarkosyl; pH < 7), to inactivate any potentially remaining prion infectivity. Resuspended pellets were transferred to a genomic DNA removal kit column (QIAGEN) and centrifuged for 20 s at 8,000$g$, RT. Flow trough was mixed with an equal volume of 70% ethanol, transferred to a RNeasy Mini kit column (QIAGEN) and prepared according to QIAGEN protocol. Total RNA was eluted with 30 µl nuclease-free water and stored at –72°C.

## RiboTag immunoprecipitation of translating mRNA from tissue homogenate

Before RiboTag purification, protein G–coated Dynabeads (PGDB; Invitrogen, Cat. no. 1009D, Lot: 00729875) were washed twice by resuspension in 1× PBS and once in PSB. IgG2b Isotype antibody (Invitrogen; Cat. no. 14473285, Lot: 2025721), diluted 1:50 in PSB, was

bound to washed beads by incubation on a MACSmix Tube Rotator (Miltenyi Biotec) at 4°C, 20 min. Isotype Ab–bound beads were resuspended in 900 µl S1, incubated rotating for 30 min, 4°C, and collected on a Millipore Magna GrIP magnetic rack (Millipore). The supernatant was incubated with 36 µl of anti-HA 12CA5 monoclonal antibody (Roche, Cat. no. 11666606001, Lot: 39746400), rotating at 4°C for 90 min. 90 µl washed PGDB were resuspended in the S1-antibody mix and incubated rotating at 4°C for 45 min. The beads were washed twice with 900 µl PSB, thrice with 900 µl HSB, and once with 900 µl EHSB. For each wash step, the beads were carefully resuspended in buffer and incubated at 300 rpm for 2–5 min (Thermomixer shaker; Eppendorf). To inactivate any remaining prion infectivity, washed beads were resuspended in 50 µl of PKB for 30 min, RT. 500 µl Qiazol (QIAGEN) were added and incubated at 300 rpm, RT, 10 min. Beads were collected on a magnetic rack and the supernatant moved to a fresh tube with 400 µl chloroform, shaken vigorously and incubated for 1 min, RT. Samples were centrifuged at 14,000$g$, RT, for 10 min, the aqueous phase moved to a fresh tube and mix with equal volume of 80% ethanol. RNA was extracted using QIAGEN RNeasy micro columns (QIAGEN) according to protocol, eluted with 30 µl of nuclease-free water and stored at –72°C. To increase the RNA yield from scarce SST$^+$ and PV$^+$ neurons, immunoprecipitation was performed using twice the volume of brain homogenate (1,800 µl S1) originating from the same mouse. Because of technical limitations, this was done by performing two parallel IPs of 900 µl for each biological replicate for SST or PV. After elution from the PGDB, these duplicate samples were stepwise added to the same RNeasy micro kit column for RNA extraction and processed as described above. No samples from biological replicates were pooled.

## Library preparation and sequencing of RiboTag and total RNA samples

Libraries were prepared at SNP&SEQ Technology platform at NGI Uppsala, using the Illumina TruSeq Stranded mRNA protocol. Quality control and quantification of RNA samples and libraries was performed using Agilent TapeStation (Agilent). Libraries were indexed and normalized, then paired end sequencing (100 bp) was performed on an Illumina NovaSeq6000 sequencer using a single S4 flow cell (Illumina). To avoid batch effects between different lanes of the flow cell, biological replicates were distributed evenly across the four lanes. Libraries of two samples (BK119.A and U84.A) failed initial sequencing and were resequenced at 150 bp PE and the same platform.

## Bioinformatic analysis

### RNAseq data availability
Code is openly available at github repository https://github.com/susannebauer/familialPrD. Raw data are deposited on GEO with accession number GSE198063.

### Alignment and mapping
Alignment was performed using the nf-core/rnaseq 3.0 analysis pipeline (53) using default settings. STAR and Salmon were used for alignment and quantification. Sequences for ERCC spike ins and

RiboTag-HA tag were included as additional Fasta file and are available on our github repository. Samples were kept if they contained >30M mapped reads and <20% ribosomal RNA reads. One *Prnp* wild-type SST⁺ sample (Z58.A) was excluded as we detected cross-contamination with *Prnp* reads containing the D178N mutation.

### Marker enrichment and PCA

To generate PCA plots, we calculated the variance for protein-coding genes based on log-scaled transcripts per million (TPM) values across either RiboTag IP or total RNA samples. Top 1% most variable genes were used for PCA with prcomp() and visualized with ggplot2 (v3.3.3). Enrichment of cell type specific marker genes in IP and total RNA samples was analyzed using gene-wise z-score of $\log_2$-transformed TPM values normalized to input (total RNA) levels using the formula by subtracting the mean TPM of total RNA sample from each sample and dividing it by the row-wise SD: $Z = (x - \text{mean(total RNA)})/\text{SD(row)}$. Heat maps were visualized using pheatmap (v1.0.12).

### Differential gene expression analysis

Differential expression analysis was performed for each cell type comparing disease and control samples (FFI versus WT and CJD versus WT) with DESeq2 (v1.30.1) (30). Salmon transcript counts were collapsed to gene levels using tximport (v1.18.0) and prefiltered to include only protein coding genes with a row-wise mean count >10. Genes with FDR-adjusted *P*-values ≤ 0.05 were considered differentially expressed, and no $\log_2$ fold change (LFC) cutoff was applied. Results from DESeq2 included several genes with extreme LFCs which occurred because of highly variable expression with zero TPM values occurring in biological replicates in control and disease groups. These genes are likely noninformative and are not visualized in dotplots in Fig 4A and B by adjusting the y-axis using the ggplot function coord_carthesian(). The original plots are shown in Table S5.

### Overrepresentation analysis

Overrepresentation of genes from gene ontology (GO) terms included in Biological Process (BP), Cellular Compartment (CC), Molecular Function (MF) collection (2021), and KEGG Mouse pathways (2019) among DEGs was performed using the enrichR R package (v3.0). Terms with adjusted *P*-value ≤ 0.01 were considered significant.

### GSEA

GSEA was performed for each cell type and disease using piano (v2.6.0) (34) for GO Biological Process (c5.go.bp.v7.4.symbols; gsea-msigdb.org) and KEGG pathways (KEGG_mouse_2019; maayanlab.cloud/Enrichr). KEGG pathways relating to tissues of different embryonic origin were removed from the list of genes sets, a detailed list of exclusion terms can be obtained from the "ReadMe.txt" file provided on our github repository. In short, enrichment analysis was performed using the runGSA() function, setting DEseq2-derived *P*-values as "geneLevelStats," LFC as "directions," "signifMethod = "geneSampling," "adjMethod = "BH" and gene set size limited to 15–500 genes. Gene set statistics for different directionality classes were calculated using six methods by setting argument for "geneSetStats" to "mean," "median," "sum," "stouffer," "reporter" or "tailStrength." Median consensus scores were calculated based on adjusted *P*-values using the integrated consensusScores() function. Terms with distinct directional FDR ≤ 0.05 in at least half of the applied gene set statistic methods were included in result tables. For Fig 5, GO terms were collapsed to parent terms using rrvigo (v1.2.0) by semantic similarity ("Resnik," threshold = 0.8).

### Topological network analysis

A co-expression weighted network was constructed by calculating the pairwise spearman $\rho$ correlation between protein-coding genes with mean TPM > 10 across samples, excluding genes with the lowest 20% variance across samples. Positive correlations (FDR ≤ 0.01) were used to construct a weighted gene co-expression network with igraph (v1.2.6) of 6,960 nodes and 616,054 edges, using spearman $\rho$ as edge weights. Community analysis was performed using the Leiden algorithm (35) (leiden v0.3.7), setting partition type to modularity vertex partition and setting the weights argument to edge weights. Clustering results were compared with a random network of equal size generated using the Erdos-Renyi G(n,M) model. Enrichment analysis for genes of the six main modules was performed using enrichR (v3.0) (54). For plotting of top-ranked gene sets (Fig S7), enriched terms (FDR ≤ 0.01) were ranked by combined score (55). igraph was used to calculate centralities. Network plots were generated using Cytoscape (v3.8.2). Hub genes were defined as top 1% nodes with highest degree centrality for each module.

### PPI network

To validate interactions of identified hub genes, we constructed a PPI network for hub genes and first neighbors using STRING interaction and functional enrichment data (using STRINGdb plug-in for Cytoscape), including PPIs with a combined confidence score ≥0.7, excluding interaction data based on text mining and database.

## Availability of Data and Materials

All code is freely and openly available at github repository https://github.com/susannebauer/familialPrD. Raw data are deposited on GEO with accession number GSE198063.

### Ethics approval and consent to participate

Mouse experiments were performed following national and local guidelines and were approved by local authorities LANUV-NRW with protocols 84-02.04.2013.A128 and 84-02.04.2013.A169.

## Supplementary Information

## Acknowledgements

The authors would like to thank the National Genomics Infrastructure (NGI) for providing assistance with library preparation and sequencing. Computations and data handling were enabled by resources provided by the

Swedish National Infrastructure for Computing (SNIC) at UPPMAX, partially funded by the Swedish Research Council through grant agreement no. 2018-05973. Support provided through the Swedish Bioinformatics Advisory Program organized by National Bioinformatics Infrastructure Sweden (NBIS) is gratefully acknowledged. The authors declare that they have no conflicts of interest. This work was supported by the Knut and Alice Wallenberg foundation and the German Center for Neurodegenerative Diseases (DZNE). We also thank Per Hammarström and Sofie Nyström for providing access to laboratory space and resources.

## Author Contributions

S Bauer: data curation, formal analysis, investigation, methodology, visualization, and writing—original draft, review, and editing.
L Dittrich: data curation, formal analysis, investigation, visualization, and methodology.
L Kaczmarczyk: investigation.
M Schleif: investigation.
R Benfeitas: conceptualization and methodology.
WS Jackson: conceptualization, investigation, project administration, and writing—original draft, review, and editing.

## Conflict of Interest Statement

The authors declare that they have no conflict of interest.

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
