## [Reviewer comments · Life Science Alliance]

Life Science Alliance

Translatome profiling in fatal familial insomnia implicates TOR signaling in somatostatin neurons

Susanne Bauer, Lars Dittrich, Lech Kaczmarczyk, Melvin Schleif, Rui Benefitas, and Walker Jackson

DOI: <https://doi.org/10.26508/lsa.202201530>

Corresponding author(s): Walker Jackson, Linköping University Hospital

Review Timeline:

Submission Date:	2022-05-23
Editorial Decision:	2022-06-27
Revision Received:	2022-08-12
Editorial Decision:	2022-09-08
Revision Received:	2022-09-09
Accepted:	2022-09-12

Scientific Editor: Novella Guidi

Transaction Report:

June 27, 2022

Re: Life Science Alliance manuscript #LSA-2022-01530-T

Dr. Walker Scot Jackson
Linköping University Hospital
Biomedical and clinical sciences
Cellbiologen Plan10
Linköping 58185
Sweden

Dear Dr. Jackson,

Thank you for submitting your manuscript entitled "Translational profiling of neuronal subtypes in fatal familial insomnia model mice reveals TOR signaling in somatostatin neurons" to Life Science Alliance. The manuscript was assessed by expert reviewers, whose comments are appended to this letter. We invite you to submit a revised manuscript addressing the Reviewer comments.

Thank you for this interesting contribution to Life Science Alliance. We are looking forward to receiving your revised manuscript.

Sincerely,

B. MANUSCRIPT ORGANIZATION AND FORMATTING:

Reviewer #1 (Comments to the Authors (Required)):

This is the first study that reports on transcriptome analysis of several neuronal cell types in mouse models of familial prion diseases. Two mouse models with E200K and D178N substitution in Prnp gene were examined at a pre-symptomatic stage of the disease. RiboTag technology was employed for examining cell type-specific transcriptome from cerebrum and cerebellum. Unexpectedly, high similarities were found in two disease models. Also unexpectedly, somatostatin neurons showed the highest number of differentially expressed genes in both disease models. Overall, this is a well done study that addresses the topic of considerable interest to the field. I recommend publication after addressing several points.

Critical points:

- 1) Page 8, line 200 "Since the mice were at a pre-symptomatic disease stage...". It is not clear from the current study whether FFI and CJD models employed in this work display well-defined clinical onset and what is the age of clinical onset for both models. Perhaps, this information is available from previous publications by this group. Nevertheless, it might be important to discuss this topic to elaborate (i) where, in preclinical diseases, the 9-month old animals are, and (ii) whether the rate of progression is similar or different in two models. While animals are examined at the same biological age, this age might not correspond to the same disease stage in these models.
- 2) Information on number of individual animals analyzed for each group (FFI, CJD and WT) with respect to total RNA and RiboTag transcriptome is missing. It is also not clear whether RNA samples were pooled together or analyzed individually for each animal.
- 3) A statement about the sex of mice analyzed in all three groups is lacking.
- 3) Presentation of Figure 2 and, specifically, panels A and C need improvement. It is not possible to discriminate between short- and long-dashed lines. Also, wake, NREM and REM are defined as gray lines, whereas only black and red lines present in the panels, which is confusing.
- 4) Pages 4-5: "...clinical abnormalities measured by automated behavioral analysis..." Information regarding parameters monitored by the automated behavioral analysis should be provided.
- 5) Page 5, lines 115-119: the sentence "Turning to the FFI..." is difficult to read, and should be revised.
- 6) Page 6, line 140. Background of WT mice has to be defined.

Reviewer #2 (Comments to the Authors (Required)):

The manuscript by Bauer et al. described the findings from their study of selective neuronal vulnerability in genetic prion disease. Using the CJD and FFI knock-in models generated in their previous studies, authors have compared translating mRNA in six neuronal types of these two animal models at a pre-clinical stage (9 months). Interestingly, they found that somatostatin neurons had the strongest response in both disease models. In addition, there are a lot of overlaps in genes and pathways as well. Based on these results, they suggested that both diseases shared a similar molecular pathways and down regulation of mTOR signaling has been suggested as the root cause. The study is well executed, results are clearly presented, and discussion is appropriate. Data presented here might be useful for researchers in the field. There are several points that need to be addressed.

1. In the study, authors separated brain into cerebellum and cerebrum. Since the most severely affected brain region is thalamus and hippocampus in FFI and CJD, respectively, is it possible that using the whole cerebrum will reduce the sensitivity of the analysis?

2. Only 4 types of neurons were analyzed. It is not entirely clear why authors think these 4 types of neurons are critical for these two genetic prion diseases. Is it possible that changes in other neuronal types or non-neuronal cells are missing?
3. In figure 3C and D, the glial genes appear to be up-regulated in total RNA, particularly in cerebellum. Other study has suggested a major role of glia in prion disease. Have authors looked into glial genes in these animals?
4. It is understandable that these mice are in a pre-clinical stage and mild changes in gene expression is expected. But at the same time, how can authors be confident that detected changes in gene expression are real and disease relevant? Is it possible to perform some type of verification to support the claim? Because all the analyses are based on this set of data,
5. Two sentences in line 220-224 are not very clear. If we are talking about the same thing, these two sentences should be combined.
6. It is not clear whether the sentence in line 227 "We also found..." is for one disease or both. In addition, the word enrichment is ambiguous and more clear words like down- or up-regulation would be helpful.
7. Authors may consider re-arranging Figure 5 in a better way. The current version is not very effective and somewhat confusing. For example, it is not very clear how authors can reach the conclusion in line 280, "...response of vGluT2 neurons in the cerebrum were more disease-specific ... "
8. The comparison was between genetic CJD and FFI, but authors failed to find differences. Instead, they have found many similarities and conclude that the neurodegenerative processes in these two diseases are similar. In addition, the key pathway suggested by authors, mTOR signaling pathway, is common in a variety of cells and is disrupted in many neurodegenerative diseases and aging. What could be the reason for two different diseases?

Reviewer #3 (Comments to the Authors (Required)):

Bauer ms.

This is an interesting study with some deep phenotyping, technically challenging using crosses to make double Tg x KI mice maintained - slightly unusual for neuroscience research - on an 129 background (putatively to remove a confounding diurnal rhythm confound present in C57B6 mice).

After an introduction to the KI mice and crosses, the paper segues into a lot of transcriptomic prose/phenomenology. While the language usage is generally fine (but see notes about abbreviations and abbreviated explanations) one would have to say that it is a dense, challenging read with little concession to the reader in introducing the diverse technologies and the expected outcomes thereof. In the same vein, the actual figures don't have numbers on them. Nor do the supplementary figures.

Fig. 2 E-J is confusing as the y axis is meant to be the proportion of (electrical ?) power at a given frequency but is labeled as if it is a statistical power. In fact, because the plot is split in two with a tiny little trace at the bottom there are two y-axes and my confusion has arisen as the vertical label can be read as one squished continuous label "p-value relative power". To this referee's eye the plots with significant p values do not look any different to those without. There is no mention of the number of animals of each genotype used to create these plots so it is difficult to know how seriously to take the data.

Fig. 3A legend, there are no grey shaded samples. Perhaps leave the shapes unfilled. The title has mild overreach, calling it a "translatome" but there are no translation experiments here, nor even outputs measured in terms of proteins, it is just a pull-down followed by some transcriptomics. The descriptor on line 142 is perhaps a closer match.

If IHC or in situ wasn't used here to confirm experimentally the assumed pattern of expression of the tagged ribosomal subunit, how do we now the tagged subunit is expressed in the mooted region specific manners in these particular mouse stocks? There is a reference, but it is only to a BioRx abstract. This is a key assumption for the experiments. The indirect answer to the question "is everything working as advertised?" is the data in Fig. 3A, I suppose. Please elaborate.

In terms of the degree of change in the "hits" and how to assess them, it would be useful to have a feel for the degree of variation found between genotypic groups versus the variation between biological replicates of the same genotype. This is an important piece of information to understand before going to town (10 figures) on data interpretation. I looked in the M&M and figure legends and eventually found these numbers in Fig. S2; perhaps they could be put into an adjusted Fig. 1.

The behavior/EEG is just done on straight KI mice, I think. There is an assumption that the whole tagging scheme is phenotypically silent, i.e., that if the behavior/EEG was done all over again on triply modified mice carrying the codon 177

variations that the result would be the same. Which brings us to the issues of the negative controls. Specifically, these are sophisticated experiments using multiple crosses so there is an issue about which WT animals from the crosses to use as the point of reference. Fig. 1A last panel implies it is merely WT mice with no Cre etc but certainly there is the ribotag genetic componentry to consider and I'm not sure if WT mice were really used for this purpose. The negative controls are mice doubly heterozygous for Cre driver and responder, according to lines 497-502, contrasting with Fig. 1A. Which is right?

Everything was backcrossed to 129S4, but how many times? It would be useful to state this, to get a feel for whether or not modifier loci are segregating in the compound mice.

I am wondering if there is a partial circularity in the logic that runs through the entire paper. The results are viewed through the lens of neurochemistry (neurotransmitters, peptide hormones), something along the lines of "aren't these somatostatin positive+ neurons interesting in prion disease?" (pages 12-14) but then again the methodology is predicated on a neurotransmitter-based view of neuroanatomy, care of the Cre driver lines. Alternative views are, for example, regional variations in proteostasis or support cells that might affect neurodegeneration caused by the mutant form of Prp could be equally interesting. A methodology that makes no assumptions about which particular activities within different brain regions are important would be the approach where these brain areas are merely dissected out prior to making RNA. Anyway it is what it is and if I have understood correctly the authors are thinking that the non-SST neurons are a control to frame the results from SST+ cells.

Another potential skew in drawing conclusions from the data would be technical, the yield of RNA recovered from the pull-downs. This is not specified (micrograms, nanograms?) but might affect how deeply one can look into the samples to find DEG's. Were RNA inputs normalized in some way? I'm asking this because there is a comment about SST and PV neuronal RNA yield (line 582).

There is little (nothing?) in the way of secondary validation of the bioinformatic hits from source tissue. e.g., the products of translation, the relevant proteins from the DEG's. The mTOR pathway highlighted in the abstract would also be a case in point. Have the authors now carried out protein analyses that would support their conclusions? While I am broadly receptive to the preliminary interpretations made here, such analyses could cement these interpretations and greatly enhance the paper.

Minor points, typos

F-CJD was renamed gCJD a number of years back to acknowledge that incomplete penetrance can lead to skipped generations and hence non-familial presentation.

Ref 16 is duplicated as ref 17.

Diagram on page 38 is for a graphic abstract?

"In our recent RML study mentioned" - jargon.

In our recent studies using inoculations with the Rocky Mountain Lab (RML) isolate of mouse-adapted scrapie prions...

"PrDs" - a borderline esoteric abbreviation that will be understood by few and is confusingly similar to PrPd used for immunoreactive signals from misfolded PrP and to PrLD used to signify low complexity unstructured domains found in yeast prions. Suggest to just say prion disease.

"model of acquired PrD" - mouse model of experimental scrapie ?

"sickest mice" - mice with the most advanced clinical disease?

The "Ribotag" handle is unfortunate as it sounds like a modified RNA nucleotide that allows a pull-down. HA-tagged ribosomal protein L22? The concept is barely introduced and expects too much of the reader. What is special about L22? Brief text could be inserted on page 3 and/or 6 to make things clearer.

GSEA is listed as a keyword, but the acronym is not spelled out until line 241.

Six neuronal types but only 4 cre drivers (line 381). Are some descriptions flipped around in lines 933-937? (Figure 1C). Please explain. Could also add the age when RNA was made from the euthanized animals to this figure

Is Fig. S1 the right size for reproduction?

We thank the editor for considering our manuscript and the reviewers for their careful reading and excellent comments and suggestions. We have addressed all of them, point by point, and believe that the revised manuscript has improved significantly, both in its scientific conclusions and its readability. In this rebuttal, all reviewer comments are in black font, while our answers are in red, and text changes to the manuscript are in blue and underlined.

Reviewer #1 (Comments to the Authors (Required)):

This is the first study that reports on transcriptome analysis of several neuronal cell types in mouse models of familial prion diseases. Two mouse models with E200K and D178N substitution in Prnp gene were examined at a pre-symptomatic stage of the disease. RiboTag technology was employed for examining cell type-specific transcriptome from cerebrum and cerebellum. Unexpectedly, high similarities were found in two disease models. Also unexpectedly, somatostatin neurons showed the highest number of differentially expressed genes in both disease models. Overall, this is a well done study that addresses the topic of considerable interest to the field. I recommend publication after addressing several points.

Critical points:

1) Page 8, line 200 "Since the mice were at a pre-symptomatic disease stage...". It is not clear from the current study whether FFI and CJD models employed in this work display well-defined clinical onset and what is the age of clinical onset for both models. Perhaps, this information is available from previous publications by this group. Nevertheless, it might be important to discuss this topic to elaborate (i) where, in preclinical diseases, the 9-month old animals are, and (ii) whether the rate of progression is similar or different in two models. While animals are examined at the same biological age, this age might not correspond to the same disease stage in these models.

We thank the reviewer for making these important points. Humans carrying the D178N mutation in context of 129M homozygosity (like the mice studied here) can have a disease onset as early as late teens to as late as late 70s. The human disease caused by the E200K mutation also has a wide-ranging disease onset and is incompletely penetrant. Like in humans, it is hard to pinpoint a specific moment when a mouse converts from healthy to diseased. Therefore, in the first subsection of the results section we described what had been previously determined about the clinical onset of disease in these mice and included the relevant references. We end this section by stating that when considering data from all methods (e.g., MRI, histopathology, behavior,) that both models show a disease onset at about 16 months of age. Then in the next section we report on our current studies of sleep and the appearance of EEG theta of mice at 21 months of age and report very mild changes, notably increases in theta that has also been reported in human prion diseases. Therefore, at 9 months, the age studied with RiboTag, there is no sign of disease. We agree that the biological age may not match the disease stage, but it is impossible to address when disease is so mild. Nonetheless, we added the following text to the discussion, starting at line 430:

Since SST⁺ neurons showed more DEGs in FFI than in CJD, and a functional analysis revealed downregulation of genes related to synaptic function and GTPase signaling, SST⁺ neurons appear to be at a more advanced disease stage in FFI than CJD brains.

2) Information on number of individual animals analyzed for each group (FFI, CJD and WT) with respect to total RNA and RiboTag transcriptome is missing. It is also not clear whether RNA samples were pooled together or analyzed individually for each animal.

The table showing the number of biological replicates for RiboTag experiments was moved from the previous Fig. S2 and the information on number of biological replicates for RiboTag and total RNA samples is now included in Fig. 3A. Total RNA was obtained from brain tissue homogenate by diverting some supernatant for this purpose, prior to performing RiboTag IP. Since these samples were expected to be largely uninformative and our purpose for them was mainly to confirm enrichment of cell type specific RNA in RiboTag IP samples, only a subset of obtained total RNA samples were sequenced in order to extract as much data as possible from the more important RiboTag IP samples by preserving most of the space on the flow cell for them. We clarified this in the legend for figure 3, as well as lines 177-178:

...we also analyzed total mRNA obtained from tissue homogenates prior to RiboTag IPs for a subset of biological replicates (Fig. 1D).

We did not pool RNA samples from different biological replicates. However, for scarce cell types (SST and PV) we doubled the volume of tissue homogenate on which IPs were performed. In these cases, the tissue homogenate was still obtained from an individual animal. Due to technical limitations, it was not possible to

simply upscale the reaction volume. Instead, we performed two IP reactions in parallel, each of 900 μ l, according to the protocol provided in the method section and used for all other samples. RNA obtained from these two duplicate reactions was pooled. We rephrased our description of this in the method section (lines 688-694).

To increase the RNA yield from scarce SST⁺ and PV⁺ neurons, immunoprecipitation was performed using twice the volume of brain homogenate (1800 μ l S1) originating from the same mouse. Due to technical limitations, we performed two parallel IPs of 900 μ l for each biological replicate for SST or PV. Following elution from the PGDB, these duplicate samples were stepwise added to the same RNeasy micro kit column for RNA extraction and processed as described above. No samples from biological replicates were pooled.

3) A statement about the sex of mice analyzed in all three groups is lacking.

Thank you for this important comment. Information on the sex of the RiboTag mice was included in the Supplementary Dataset S1. To make this information more prominent, we added a table as new supplementary figure Fig. S2A. Information about the sex of mice used for EEG studies was missing and has now been added to the methods section. The break down included: FFI – 15 (11 female), WT – 14 (9 female), CJD – 8 (4 female).

3) Presentation of Figure 2 and, specifically, panels A and C need improvement. It is not possible to discriminate between short- and long-dashed lines. Also, wake, NREM and REM are defined as gray lines, whereas only black and red lines present in the panels, which is confusing.

Using neutral colors such as grey is typically done to convey a pattern that is repeated for multiple groups. However, the point that the dashed and dotted lines were hard to discriminate is very important. We therefore restyled the figure to accommodate both concerns. In the new figure the dots and dashes are more easily discernible, and the key does not include a neutral-colored symbol. We took this opportunity to make an additional change to keep with the color theme throughout the manuscript: now the control group is shown in black, FFI in red, and CJD in blue.

4) Pages 4-5: "...clinical abnormalities measured by automated behavioral analysis..." Information regarding parameters monitored by the automated behavioral analysis should be provided. We added the clause "used to measure multiple activities of mice (e.g., roaming, grooming, distance traveled, rest, etc.) in home cages" to generate the following sentence (lines 102-107):

Automated mouse behavioral analysis used to measure multiple activities of mice (e.g., roaming, grooming, distance traveled, rest, etc.) in home cages indicated sleep was fragmented and core body temperature measurements suggested FFI mice had impaired sleep regulation at this age (Steele *et al.*, 2007), but electroencephalography (EEG) measurements were not attempted then due to biosafety constraints.

5) Page 5, lines 115-119: the sentence "Turning to the FFI..." is difficult to read, and should be revised.

This sentence was changed to (lines 118-122):
Since we previously observed that behavioral activity in FFI mice was only mildly affected at 16 months of age, which is likely a result of only mildly diminished neural health at that timepoint, we sought to increase the possibility of detecting EEG abnormalities by studying old mice at approximately 21 months of age (mean = 20.8, SD = 2.3).

6) Page 6, line 140. Background of WT mice has to be defined.

Thank you for this important point. We included these details in the methods section but now realize we should also include them more prominently in the main text.

Starting at line 167 we add this sentence:

Since the commonly used C57Bl/6 strain is hyperactive at night (Kaczmarczyk *et al.*, 2021), and we worried this would introduce unwanted changes to gene expression patterns, we used the calmer 129S4 strain for all mice in this report (details in methods).

Reviewer #2 (Comments to the Authors (Required)):

The manuscript by Bauer et al. described the findings from their study of selective neuronal vulnerability in genetic prion disease. Using the CJD and FFI knock-in models generated in their previous studies, authors have compared translating mRNA in six neuronal types of these two animal models at a pre-clinical stage (9 months). Interestingly, they found that somatostatin neurons had the strongest response in both disease models. In addition, there are a lot of overlaps in genes and pathways as well. Based on these results, they suggested that both diseases shared a similar molecular pathways and down regulation of mTOR signaling has been suggested as the root cause. The study is well executed, results are clearly presented, and discussion is appropriate. Data presented here might be useful for researchers in the field. There are several points that need to be addressed.

1. In the study, authors separated brain into cerebellum and cerebrum. Since the most severely affected brain region is thalamus and hippocampus in FFI and CJD, respectively, is it possible that using the whole cerebrum will reduce the sensitivity of the analysis?

The reviewer brings up a great question that we have struggled with for years.

Since the hippocampus and thalamus are located in the cerebrum and are especially enriched in glutamatergic neurons, we expected that vGluT2/cerebrum samples would have the most differentially expressed genes, but they did not. Moreover, these regions have very few SST neurons, which had the most changes. Therefore, we are glad we did not focus on these regions at the expense of others. Nonetheless, we do see that in this sample type (vGluT2/cerebrum) there are differences between diseases when considering the gene set enrichment analysis (Fig 5). We originally avoided focusing on these two regions because it is hard to dissect out the thalamus cleanly, and without it, a study of the hippocampus alone was less interesting. Therefore, we focused on the cerebellum because it is easy to dissect out accurately and precisely and was affected in both diseases. But to point out this important point we added a limitations section to the discussion that includes the following text (lines 537 -544):

Finally, we intended to address the topic of selective vulnerability, but practical matters constrained us to study a limited number of brain regions and cell types. One constraint was that the regions selectively targeted in FFI and CJD mice, the thalamus and hippocampus, respectively, are difficult to dissect precisely (especially the thalamus) and were therefore studied together as components of the cerebrum samples. This complex mixture of regions likely obscured differences that exist. Nonetheless, with this approach we determined that SST neurons are highly changed, and since they are scarce in the thalamus and hippocampus, we would have missed this discovery had we focused on those regions.

2. Only 4 types of neurons were analyzed. It is not entirely clear why authors think these 4 types of neurons are critical for these two genetic prion diseases. Is it possible that changes in other neuronal types or non-neuronal cells are missing?

We agree this is also an important question. First, with respect to neurons, the combination of Cre lines we employed cover most neurons in the brain. PV neurons have been reported to be most affected in most prion diseases, in humans and animals and were therefore included in this study. GABAergic SST+ neurons are interesting because in most brain areas they do not overlap with PV neurons, thus allowing a clear separation between these neuronal subtypes in our analysis. Additionally, while SST neurons had not previously been reported to show early response in prion disease, early vulnerability of hippocampal SST+ interneurons have been reported in APP/PS1 mice (Schmid et. al. 2016). Gad2 and vGluT2-expressing neurons were added as additional complementary neuron types, as described in the original introduction. Additionally, by using the Gad2 and vGluT2 Cre lines, we were also able to target the principal neuronal populations in the cerebellum, a region affected in both diseases, broadly separated by granular and molecular layer.

In respect to non-neuronal cells, astrocytes and microglia came to our mind. We previously used an astrocyte specific Cre line for our RML studies and attempted to use it for this study but ran into breeding issues and could not recover enough animals for all genotypes to include in this study. Prior to starting these studies, in collaboration with Annett Halle at DZNE, we attempted to activate the RiboTag gene in microglia using two microglia-targeting cre lines (LysM and CD11b) and neither activated RiboTag in more than 10% of microglia. The Cx3Cr1-CreERT2 line that is now widely used was not published when we began our study.

Finally, other cell types could have been studied, such as subtypes of blood vessel cells or oligodendrocytes. We think these in addition to studies of glia are worth doing, but now in the context of Tagger mice (Kaczmarczyk, et al., 2019) which enables the study of additional layers of gene regulation. To bring this important point into the manuscript we added the following text to the limitations section of the discussion (545-553):

The number of cell types to study was constrained because the intercross to obtain experimental mice required mice that were double homozygous for Prnp and either Cre or RiboTag genes, requiring several rounds of breeding and lots of mice and mouse room space. Therefore, we focused on a limited set of neurons and changes to non-neuronal cells remain unknown. Considering the results reported here for neurons of FFI and CJD mice, and from a separate study of an acquired prion disease model studying neurons and astrocytes [18], future studies of FFI and CJD mice examining more cell types, including astrocytes, microglia, oligodendrocytes, and neurovascular cells would likely provide a deeper understanding and bring additional value to the current data.

3. In figure 3C and D, the glial genes appear to be up-regulated in total RNA, particularly in cerebellum. Other study has suggested a major role of glia in prion disease. Have authors looked into glial genes in these animals?

Thank you for this very important point.

The apparent upregulation of astrocyte and microglia markers in total RNA samples was a result from how the row-wise z-score was calculated. Initially, we calculated the z-score for each gene across all samples. This meant calculations were done across both, cell type-specific RiboTag IP samples -which we would not expect to show high expression levels of glial genes as only neurons were targeted- and a smaller number of total RNA samples, which contain mRNA from all cell types. As a result of this approach, total RNA samples showed an apparent upregulation of glial genes compared to the mean calculated across all samples. Following internal discussions, we decided this approach to z-score calculation misrepresents the data, since expression levels of total RNA should be considered a baseline expression level to which enrichment (by RiboTag) is compared. We therefore recalculated the z-scores, this time normalizing RiboTag samples to total RNA input levels, a method we used previously (Kaczmarczyk *et al.*, 2019). To achieve this, we calculated the mean TPM value across total RNA samples. The mean(total RNA) was then subtracted from the sample value and divided by the row-wise standard deviation to obtain a normalized z-score. This was updated in the method section (lines 724-727) and the revised figure legend for figure 3 (lines 1089-1106). We also revised the description in the result section to emphasize that these are normalized values (lines 190-194).

This was apparent through comparisons of expression of cell type marker genes in RiboTag IP samples normalized to total RNA expression levels, which revealed the expected relative enrichment of general GABAergic and glutamatergic neuronal marker genes in respective RiboTag IP samples (Fig. 3D and E).

Recalculated values are displayed in Figure 3 D+E and more clearly show a depletion of glial genes in RiboTag samples obtained from cerebrum and cerebellum when compared to baseline expression levels in total RNA samples. The re-analyzed results do not change the overall interpretation of the data as presented in the result section.

As mentioned in our reply to question 2 above, we were thwarted from studying glia in the current study but we now have new Cre and Cre responder lines that enable us to do so in the future.

4. It is understandable that these mice are in a pre-clinical stage and mild changes in gene expression is expected. But at the same time, how can authors be confident that detected changes in gene expression are real and disease relevant? Is it possible to perform some type of verification to support the claim? Because all the analyses are based on this set of data,

Thank you for this comment.

The purpose of this project is to obtain candidate therapeutic targets to study in more detail. As such, follow-up studies are not typically done with these types of methods upon the initial presentation of the "screen" because a lot more work is needed. The main problem is that with small changes in specific cells, conventional methods like western blotting can't be used. Similarly, immunohistochemistry is difficult to use since it is not very quantitative, especially when most cells in the tissue have not changed their expression of the test gene. Furthermore, in our case, RNA in situ hybridization can't be done on samples treated with formic acid because the RNA gets hydrolyzed, and samples not treated with formic acid cannot be used on specialized equipment that is not in a biocontainment lab.

Nonetheless, we are confident in the results for multiple reasons and have added the following text to the limitations section (lines 522-530):

Nonetheless, there are multiple reasons to be confident in the overall results. First, we have a proper replicate size and the data were analyzed with multiple, distinct bioinformatics tools giving the same overall result. Second, a study of Huntington's disease using bacTRAP (a method analogous to RiboTag) showed the method is reliable since they found similar results with single cell analysis [47]. Third, we have used this method to study Huntington's disease in parallel with the current study and acquired prion disease samples in a separate experiment, and both times found different cell types changing unique pathways, indicating that the results are disease model-dependent, as expected.

5. Two sentences in line 220-224 are not very clear. If we are talking about the same thing, these two sentences should be combined.

Thank you for pointing this out. The lines have been revised (lines 248-252):

Cytoskeleton and cell adhesion-related terms were enriched among both up- and downregulated DEGs in FFI SST+ neurons (up: "myelination", "actin-binding", "focal adhesion" and "cell-substrate junction"; down: "processes related to neurite morphogenesis and organization", "microtubule binding" and "motor activity", synaptic plasticity and ion-channels or receptor components) (Fig. S5B).

6. It is not clear whether the sentence in line 227 "We also found..." is for one disease or both. In addition, the word enrichment is ambiguous and more clear words like down- or up-regulation would be helpful.

Thank you for this helpful comment. The lines have been revised (lines 252-255):

GTPase activity-related genes, such as activators of Rho-family GTPases (*Arhgap32,35,44*), Rho guanine nucleotide exchange factors (GEFs) (*Als2, Agap2, Trio, Dock4*), and downstream effectors (*Cdc42bpa, Rock2*) were also overrepresented among downregulated DEGs in FFI SST+ neurons.

7. Authors may consider re-arranging Figure 5 in a better way. The current version is not very effective and somewhat confusing. For example, it is not very clear how authors can reach the conclusion in line 280, "...response of vGluT2 neurons in the cerebrum were more disease-specific ..."

Thank you for this comment; it has motivated us to change it significantly. In the original figure 5, we summarized related GO terms found to be significantly up- or downregulated in our GSEA results across diseases and cell types. While this method has some shortcomings, it enabled us to display whether directional changes related to certain terms is unique to a specific disease model, occurred in both diseases but with opposite directionalities, or occurred in both diseases with the same directionality. This was used to estimate whether neurons of a specific cell type showed a broadly similar response in both disease models, or more disease-specific responses between models. We decided on the visualization used in Figure 5 as a space-efficient way to display a large amount of data that should also easily communicate differences between the diseases and cell types. From that original figure we had the impression that vGluT2 neurons were more likely than other neurons to have different responses because they had the most terms that changed in opposite directions. However, to make it easier for the reader to find discussed pathways, we changed the figure to reduce the number of terms to those discussed in the manuscript or those we deemed most relevant in the context of our other results, and we include a panel that summarizes the results to more clearly show this effect. The original figure 5 has been moved to the supplement. We also provide the full results table as supplementary dataset 4. The text related to this section was heavily edited and therefore too large to include in this response letter; please see the manuscript directly.

8. The comparison was between genetic CJD and FFI, but authors failed to find differences. Instead, they have found many similarities and conclude that the neurodegenerative processes in these two diseases are similar. In addition, the key pathway suggested by authors, mTOR signaling pathway, is common in a variety of cells and is disrupted in many neurodegenerative diseases and aging. What could be the reason for two different diseases?

We expected very little overall similarity in gene expression changes between our CJD and FFI models and based on results from our study of acquired prion disease, we did not expect to see a response in SST neurons. Therefore, the results for SST neurons were surprising us and given recent reports of early dysfunction of SST⁺ neurons in other neurodegenerative diseases and the involvement of SST⁺ neurons in processes that are commonly affected in NDs and aging, such as sleep, we thought it was important to highlight this observation. We also detected numerous differences between the diseases and thanks to the reviewers comment we have emphasized these differences more clearly. But first a little more context for the mTOR connection.

Despite mTOR signaling changes being a widespread phenomenon in a variety of diseases and cell types, cell type specific differences to mTORC1 disruption have been demonstrated. A recent study even reported that increasing mTORC1 activity by Tsc1-KO changed electrophysiological properties and dendritic morphology of striatal GABAergic medium spiny neurons (MSN) expressing the type 1 dopamine receptor, but not of MSNs expressing the type 2 dopamine receptor (Benthall et al, 2018), two very closely related and spatially close neuronal subtypes. It was also recently demonstrated, that Tsc1-mediated mTOR inhibition plays a role in maintaining electrophysiological properties and cell identity in a subset of cortical SST interneurons (Malik et al 2019). These results indicate that despite mTOR disruption being commonly observed in a variety of diseases as well as during the natural aging process, disruption of mTOR may have cell type-specific effects and making it conceivable that mTORC1 disruption in SST neurons may have impact on SST⁺ function in our models of genetic prion disease.

Additionally, RiboTag analysis of SST⁺ neurons in RML (referenced in the manuscript) and in HD KI-mice (ongoing study) revealed very little (practically no) response in SST neurons at all and we did not observe similar changes in mTOR signaling pathway regulation in any cell type for either study. To us, this suggests that the changes observed in SST neurons are not a general response of this cell type to neurodegenerative disease or aging but seemingly constitutes a shared feature of genetic forms of FFI and CJD.

Getting back to the reviewer's question "why different diseases?" we do not claim that the changes observed in SST⁺ neurons are the underlying cause for the different disease manifestation between CJD and FFI. Instead, it likely comes from the other cell types where we see differences. We think the changes we made to the manuscript related to question 7, both in figure 5 and the related text, have also brought clarity to this issue. We also added the following text to the discussion to address this important point (lines 571-583):

The largest differences between FFI and CJD were in vGluT2 neurons of the cerebellum, while vGluT2 cells of the cerebrum showed a mixed response. This mixed response may be related to the selective vulnerability reported previously since each disease causes neuropathological changes in different brain regions, thalamus and hippocampus, both enriched in glutamatergic (vGluT2⁺) neurons. This is because in the cerebellum nearly all glutamatergic neurons are granule cells, a very homogeneous cell type and changes will not be obscured by non-responding cells. Since the cerebrum was a mixture of many regions, some affected and many not, the mixed response we observed for vGluT2⁺ cerebrum samples would be expected. Similarly, GSEA of PV⁺ neurons demonstrated a mix of terms, potentially reflecting the difference in vulnerability reported previously. A unifying

explanation of the causes of selective vulnerability remains elusive but continued experimentation with methods like RiboTag or Tagger (47) may help to eventually solve this mystery.

Reviewer #3 (Comments to the Authors (Required)):

Bauer ms.

This is an interesting study with some deep phenotyping, technically challenging using crosses to make double Tg x KI mice maintained - slightly unusual for neuroscience research - on an 129 background (putatively to remove a confounding diurnal rhythm confound present in C57B6 mice).

After an introduction to the KI mice and crosses, the paper segues into a lot of transcriptomic prose/phenomenology. While the language usage is generally fine (but see notes about abbreviations and abbreviated explanations) one would have to say that it is a dense, challenging read with little concession to the reader in introducing the diverse technologies and the expected outcomes thereof. In the same vein, the actual figures don't have numbers on them. Nor do the supplementary figures.

Historically, and still today, many journals prefer that figures not be labeled with a figure number in the figure itself, so we always submit figures this way.

Fig. 2 E-J is confusing as the y axis is meant to be the proportion of (electrical ?) power at a given frequency but is labeled as if it is a statistical power. In fact, because the plot is split in two with a tiny little trace at the bottom there are two y-axes and my confusion has arisen as the vertical label can be read as one squished continuous label "p-value relative power". To this referee's eye the plots with significant p values do not look any different to those without. There is no mention of the number of animals of each genotype used to create these plots so it is difficult to know how seriously to take the data.

Thank you for this comment. This figure received critical suggestions from the other reviewers as well, so it has been reworked in several ways. The full spectra (0-50) are shown since it is in the theta region (5-10 Hz) where the differences are most abundant (i.e., areas with no changes are also shown). To make it easier to see the theta regions, a zoom in for each was added to the corresponding panel. The number of mice included 15 FFI, 14 WT and 8 CJD. It was originally described in the methods, but to aid the reader we have now included this information in the figure legend as well.

Fig. 3A legend, there are no grey shaded samples. Perhaps leave the shapes unfilled.

We used grey-filled shapes rather than outlines to increase legibility. Grey was chosen as a neutral color to avoid confusion with color-coded cell types shown above. To avoid potential misinterpretation, we followed the suggestion to use shape outlines in the legend for Fig. 3 (now panels B+C).

The title has mild overreach, calling it a "translatome" but there are no translation experiments here, nor even outputs measured in terms of proteins, it is just a pull-down followed by some transcriptomics. The descriptor on line 142 is perhaps a closer match.

The term "translatome" is not our invention and is commonly used to refer to the collection of ribosome-bound mRNAs which may be obtained through different methods such as RiboTag, TRAP, ribosome or polysome profiling. Furthermore, tRNAs, miRNAs, lncRNAs, lincRNAs, rRNAs, snRNAs, snoRNAs and mRNAs not associated with ribosomes, such as those in granules, are all part of the transcriptome but are not what we study here. We prefer this term to the proposed alternative of "cell type-specific mRNA" as it is more concise and highlights that the isolated mRNA is translating mRNA.

If IHC or in situ wasn't used here to confirm experimentally the assumed pattern of expression of the tagged ribosomal subunit, how do we know the tagged subunit is expressed in the mooted region specific manners in these particular mouse stocks? There is a reference, but it is only to a BioRx abstract. This is a key assumption for the experiments. The indirect answer to the question "is everything working as advertised?" is the data in Fig. 3A, I suppose. Please elaborate.

The reference was a full manuscript on BioRxiv and is our manuscript using the same RiboTag and Cre mouse lines to study the RML mouse model of acquired prion disease. It has since been accepted for publication in a peer reviewed journal and the reference will be updated once available. In that manuscript we perform extensive immunofluorescence and co-localization analyses to determine where the RiboTag gene is activated, and yes, it worked as advertised. In addition to those data, the data in figure 3B-E also show that it is working as advertised.

In terms of the degree of change in the "hits" and how to assess them, it would be useful to have a feel for the

degree of variation found between genotypic groups versus the variation between biological replicates of the same genotype. This is an important piece of information to understand before going to town (10 figures) on data interpretation. I looked in the M&M and figure legends and eventually found these numbers in Fig. S2; perhaps they could be put into an adjusted Fig. 1.

Thank you for this suggestion, we moved the panel from former Fig. S2 to the main figures and modified the figure to more clearly show variation at a cell type level. However, since this panel concerns only data from our RNAseq study, we thought it would be more appropriate to include as panel A in Figure 3.

The behavior/EEG is just done on straight KI mice, I think. There is an assumption that the whole tagging scheme is phenotypically silent, i.e., that if the behavior/EEG was done all over again on triply modified mice carrying the codon 177 variations that the result would be the same. Which brings us to the issues of the negative controls. Specifically, these are sophisticated experiments using multiple crosses so there is an issue about which WT animals from the crosses to use as the point of reference. Fig. 1A last panel implies it is merely WT mice with no Cre etc but certainly there is the ribotag genetic componentry to consider and I'm not sure if WT mice were really used for this purpose. The negative controls are mice doubly heterozygous for Cre driver and responder, according to lines 497-502, contrasting with Fig. 1A. Which is right?

Thank you for pointing this out. We tried to make the details clear without taking up too much text in the manuscript, but these are important details. The EEG/sleep study was done on singly transgenic mice, where in addition to FFI and CJD, the WT controls were also knock-ins. There is a possibility that the additional Cre and RiboTag transgenes could have affected EEG/sleep results but it was not feasible to do all the possible combinations with RiboTag and Cres and Prnp genotypes.

For the RiboTag experiments, control mice expressed unmodified WT PrP and the Rpl22/Cre component. WT knock-in mice were not used for the following practical reasons. The RiboTag experiments required breeding mice to be homozygous for Cre and RiboTag. In the case of FFI, for example, this required creating five double homozygous lines (FFI+Cre x 4, FFI+RiboTag x 1). These 5 lines were then crossed to make 4 lines of experimental animals. To do this for CJD required another 9 (5+4) lines. There was simply no more capacity to make 5 lines of double homozygotes also for 3F4-WT mice. The section in the methods was reordered to improve readability. We think this level of detail is best kept in the methods section (Lines 587-606).

Figure 1A was meant to primarily highlight the differences in regional pathologies of the two disease models. We recognize that the panel showing the control brain without clear labeling of the genotype may be confusing to readers. We therefore removed the panel in Figure 1A. We have also highlighted that wild type mice used for the RNAseq study have unmodified wild type *Prnp* in the main text (line 167) and the method section.

Everything was backcrossed to 129S4, but how many times? It would be useful to state this, to get a feel for whether or not modifier loci are segregating in the compound mice.

Thank you for asking this. We think it is important and have spent great effort in working on this, but assumed most people do not care, so we did not make these details prominent. These details can be found in the paper we referenced for the interested reader. In brief, all lines were back crossed at least 8 generations, some much further. The backcrossing of Cre and RiboTag lines was guided by SNP analysis and the final lines were analyzed with a genome wide SNP analysis that included 347 SNPs that discriminate between the starting and final backgrounds, revealing the genetic background for each was at least 99.8 % 129S4. FFI, CJD and WT mice were not subjected to SNP analyses but were back crossed for nine generations and should therefore be at least 99.5% 129S4. We have added these details to the methods:

(Line 590-593)

All lines were backcrossed to the 129S4 background for at least 8 generations. Cre and RiboTag mice were at least 99.8% 129S4 (details in [25]), and the FFI, CJD and WT mice were back-crossed nine generations and therefore approximately 99.6% 129S4.

Regarding the possibility of modifiers, it is impossible to rule it out in our experiments, but we think it is likely not an issue for a few reasons. We have done similar studies with the RML model of acquired prion disease and a knock-in model of Huntington's disease and neither showed changes in SST neurons. Therefore, modifiers to get the SST differences would have to be a result of an interaction of modifiers at the Prnp and SST loci. Second since the SST locus was heterozygous, it would need to act dominantly. Third, it is in both FFI and CJD so must be very closely linked. However, since it is still possible, we have added a limitations section to the discussion which includes the following text (lines 505-518):

First, the mice were backcrossed onto the 129S4 background which could result in linked genetic modifiers impacting our results. Indeed, the signal regulatory protein alpha (Sirpa) is only 2 million base pairs away from Prnp and was reported to drive a phenotype in Prnp knock-out mice [46] that was previously attributed to Prnp. This is of little concern for the EEG and sleep studies since the control and mutant mice carried knock-in alleles engineered the same way, but the RiboTag study employed control mice carrying an unmodified wildtype allele.

Although this is difficult to dismiss in our experiments, we think it is not a problem since both, the strain of embryonic stem cells used to make the Prnp knock-in mouse lines and the mouse strain they were backcrossed to, are 129 substrains. Indeed, Sirpa sequences in the mutant and control mice are identical. Furthermore, the DEGs were distributed randomly across the genome and were not enriched for being located within 40 million base pairs of Prnp, the maximum amount of flanking genome we estimate could remain in the knock-in mice.

I am wondering if there is a partial circularity in the logic that runs through the entire paper. The results are viewed through the lens of neurochemistry (neurotransmitters, peptide hormones), something along the lines of "aren't these somatostatin positive+ neurons interesting in prion disease?" (pages 12-14) but then again the methodology is predicated on a neurotransmitter-based view of neuroanatomy, care of the Cre driver lines. Alternative views are, for example, regional variations in proteostasis or support cells that might affect neurodegeneration caused by the mutant form of Prp could be equally interesting. A methodology that makes no assumptions about which particular activities within different brain regions are important would be the approach where these brain areas are merely dissected out prior to making RNA. Anyway it is what it is and if I have understood correctly the authors are thinking that the non-SST neurons are a control to frame the results from SST+ cells.

Yes, the non-SST neurons can be seen as a control for the SST neurons

Another potential skew in drawing conclusions from the data would be technical, the yield of RNA recovered from the pull-downs. This is not specified (micrograms, nanograms?) but might affect how deeply one can look into the samples to find DEG's.

Thank you for these points, it is possible readers would have the same question. Since SST and PV neurons are less abundant than vGluT2 and Gad2 neurons, we expected to capture less mRNA, which is what we observed. This simply means that the method is working as expected. Due to the scarcity of these two cell types, we increased the volume of tissue homogenate used for the RiboTag immunoprecipitation (we explain this in detail in our answer to Question #2 from reviewer 1). This allowed us to obtain sufficient RNA for downstream library preparation and an abundance of RNA was sent to the sequencing facility. The average yield for RiboTag IPs of SST neurons, the least abundant cell type investigated, was 76 ng. Importantly, we did not find any significant differences in yields for a specific cell type between the three genotypes. We added this information to the result section and included a panel in figure S2 showing yields by cell type and genotype. Additionally, this information was added to the sample details provided in Supplementary Dataset 1.

Lines(179-183)

As was expected, yields varied greatly based on the abundance of the targeted neurons. Average yields from RiboTag IPs ranged from 76 ng of RNA from SST samples (least abundant cell type; SD: 25 ng) and 910 ng from cerebral vGluT2 samples (most abundant cell type; SD = 230 ng). However, we found no significant differences in RiboTag IP RNA yields between different genotypes (Fig. S2B).

Were RNA inputs normalized in some way? I'm asking this because there is a comment about SST and PV neuronal RNA yield (line 582).

The point of that statement was that the yields should be lower for less abundant cell types, and that is what we observed. To clarify this point, we added the text described in the previous response, and again here (line 179):

As was expected, yields varied greatly based on the abundance of the targeted neurons.

The short answer to this question is yes, the RNA input for sequencing was normalized. As stated in the methods, library preparations and sequencing were performed at the SNP&SEQ facility in Uppsala, Sweden according to specifications of the Illumina TruSeq stranded mRNA library preparation protocol. Following QC and quantification, indexed libraries were normalized according to protocol specification and samples for the different S4 flow cell lanes were pooled for sequencing. We added the requested information on RNA yields from RiboTag immunoprecipitations to the sample sheet provided as Supplementary Dataset 1 and expanded the explanation in the method section (lines 700-703):

Libraries were indexed and normalized, then paired end sequencing (100 bp) was performed on an Illumina NovaSeq6000 sequencer using a S4 flow cell (Illumina, San Diego CA). To avoid batch effects between different lanes of the flow cell, biological replicates were distributed evenly across the four lanes.

There is little (nothing?) in the way of secondary validation of the bioinformatic hits from source tissue. e.g., the products of translation, the relevant proteins from the DEG's. The mTOR pathway highlighted in the abstract would also be a case in point. Have the authors now carried out protein analyses that would support their conclusions? While I am broadly receptive to the preliminary interpretations made here, such analyses could cement these interpretations and greatly enhance the paper.

We agree that it would be nice to confirm these results with alternative methods. However, this is typically not done in other papers using these methods (RiboTag or TRAP) to study neurodegenerative diseases.

During our work with RML, which carried over to these studies, we encountered two problems. First, the formic acid treatment needed to render prion samples safe for handling makes many antibodies bind poorly. Second, most cells will express these proteins at normal levels. This causes the problem that most antibody is absorbed by the antigens in other neurons, compromising our ability to titrate antibodies to a concentration where differences could be revealed. Western blotting was not attempted for the same reason, the changes in a subset of cells would have been masked by the steady expression levels in all the other brain cells. To validate our results, we employed a series of bioinformatics analyses that indicated that the results were not random. This experiment was essentially a screen for new genes that might be used to modify disease. Future studies are envisioned to manipulate genes with Cre-dependent adeno-associated viral vectors encoding cDNAs to increase, or shRNAs to reduce, expression of genes of interest to test the impact on disease, but this requires several years to establish.

Minor points, typos

F-CJD was renamed gCJD a number of years back to acknowledge that incomplete penetrance can lead to skipped generations and hence non-familial presentation.

We see in the newly published literature both names used, but have changed it to be genetic CJD.

Ref 16 is duplicated as ref 17.

Thank you for bringing this to our attention. This has been corrected

Diagram on page 38 is for a graphic abstract?

On pg 39 is figure 1, which is in a sense a graphical description of the experimental plan

"In our recent RML study mentioned" - jargon.

In our recent studies using inoculations with the Rocky Mountain Lab (RML) isolate of mouse-adapted scrapie prions...

The sentence was changed to

In our recent study on RML-infected mice mentioned above [18]...

"PrDs" - a borderline esoteric abbreviation that will be understood by few and is confusingly similar to PrPd used for immunoreactive signals from misfolded PrP and to PrLD used to signify low complexity unstructured domains found in yeast prions. Suggest to just say prion disease.

It is normal to call Huntington's disease HD and Alzheimer's disease AD. It's also common to call Parkinson's disease PD, which makes it difficult to use PD for prion disease so many researchers, not just us, use the abbreviation PrD for prion disease. We have also used this abbreviation in a parallel manuscript on a RiboTag analysis of the RML model of prion disease which has been accepted for publication. Since we anticipate both manuscripts will be read in parallel, we respectfully wish to keep this abbreviation in place to keep the manuscripts using similar terms. However, in the future we will try to avoid the term PrD in our writings.

"model of acquired PrD" - mouse model of experimental scrapie ?

There are roughly three types of prion disease: genetic, sporadic, and acquired, this sentence is to emphasize that RML is a model of acquired prion disease and is different from genetic prion disease

"sickest mice" - mice with the most advanced clinical disease?

This has been changed as suggested

The "RiboTag" handle is unfortunate as it sounds like a modified RNA nucleotide that allows a pull-down. HA-tagged ribosomal protein L22? The concept is barely introduced and expects too much of the reader. What is special about L22? Brief text could be inserted on page 3 and/or 6 to make things clearer.

Thank you for this comment. This method was published with the name "RiboTag" 13 years ago by another group and is cited many times so we don't want to rename it. As it is becoming an established method, one of our co-authors thought we should exclude the schematic showing the method. Nonetheless, Figure 1 includes a schematic of the full process. We have now also added some text that explains how it works:

Line 151 onwards:

The RiboTag transgene is embedded into the large subunit ribosomal protein (Rpl22) whereby, following activation with a cell type-specific Cre recombinase, a version of Rpl22 fused to the hemagglutinin (HA) antibody

epitope is expressed, and HA-tagged ribosomes can be immunoprecipitated (Fig 1D). As a part of the large ribosomal subunit, Rpl22 will only associate with mRNAs when part of a complete, functional ribosome and thus RiboTag-captured mRNAs represent the translome. Importantly, mRNAs associated with only the small subunit, as well as those not associated with ribosomes at all, are not captured by RiboTag and therefore are excluded from our analysis.

Regarding the “modified RNA nucleotide that allows a pull-down” we do this with TU-tagging in the Tagger mice (Kaczmarczyk, et al., 2019).

GSEA is listed as a keyword, but the acronym is not spelled out until line 241.

GSEA, among others, was removed to adhere to recommended six keywords in LAS publishing guidelines.

Six neuronal types but only 4 cre drivers (line 381). Are some descriptions flipped around in lines 933-937? (Figure 1C). Please explain.

Thank you for this comment, we see that it can be confusing. We used four Cre driver lines but for vGluT2 and Gad2 neurons we isolated RNA from cerebellum and cerebrum separately, giving 4 cell types in the cerebrum + 2 cell types in the cerebellum = 6 studied cell types. vGluT2-expressing neurons in the cerebellum consist predominantly of granular neurons, constituting a homogenous population quite different from vGluT2 neurons in the cerebrum, which are often large pyramidal shaped neurons. Similarly, Gad2-expressing neurons in the cerebellum include Purkinje and a few cell types in the molecular layer which are also quite different in shape and activity from Gad2 neurons in the cerebrum. This, in addition to reported cerebellar neuropathology in both prion diseases motivated us to analyze cerebellum and cerebrum separately. Given these major differences in glutamatergic and GABAergic populations between cerebrum and cerebellum, we describe these as separate cell types. We think this nomenclature is justified, considering that there are no strict conventions about when neuronal populations may be considered as separate cell types. However, we understand that experimental setup and how we used four Cre driver lines to analyze six cell types must be explained better. We therefore adapted panel Fig. 1 C and D to show more clearly which cell types were analyzed in which region and highlight that cerebellum and cerebrum were processed and analyzed separately. We additionally made smaller changes throughout the main text (please see below), to clarify the experimental setup.

Legend for Figure 1, panels C and D (lines 1053-1069)

(C) For cell type-specific translome analysis, four Cre-driver lines were used to target neuronal subtypes in the cerebellum and cerebrum....

Lines 74-78:

In this study, we analyzed translome changes in vGluT2+ glutamatergic neurons (excitatory) and Gad2+ GABAergic (generally inhibitory) neurons in the cerebrum and cerebellum, as well as in cerebral GABAergic subpopulations expressing neuropeptides parvalbumin (PV) or somatostatin (SST), which are non-overlapping in most brain regions.

Lines 406-408:

Here we report cell type-specific responses in knock-in mouse models of two genetic PrDs at a pre-symptomatic stage, by translomic analysis of vGluT2, Gad2, PV, and SST expressing neurons in the cerebrum, and vGluT2 and Gad2 expressing neurons of the cerebellum.

Could also add the age when RNA was made from the euthanized animals to this figure

The average age of mice was previously provided in the method and results section and has now also been added to panel 1D and in the respective figure legend, as was suggested. Exact ages in weeks for each mouse included in our NGS study were already provided in “Supplementary Dataset S1”.

Is Fig. S1 the right size for reproduction?

This figure has been remade and is now the correct size

September 8, 2022

RE: Life Science Alliance Manuscript #LSA-2022-01530-TR

Dr. Walker Scot Jackson
Linköping University Hospital
Biomedical and clinical sciences
Cellbiologen Plan10
Linköping 58185
Sweden

Dear Dr. Jackson,

Thank you for submitting your revised manuscript entitled "Translatome profiling in fatal familial insomnia implicates TOR signaling in somatostatin neurons". We would be happy to publish your paper in Life Science Alliance pending final revisions necessary to meet our formatting guidelines.

- please use the [10 author names, et al.] format in your references (i.e. limit the author names to the first 10)
- please add a figure callout for Figure 2E and 2I and Figure S6 to your main manuscript text
- you refer to supp. Datasets S1-S6 in Manuscript; please upload these files as they are missing

A. FINAL FILES:

B. MANUSCRIPT ORGANIZATION AND FORMATTING:

**Submission of a paper that does not conform to Life Science Alliance guidelines will delay the acceptance of your

manuscript.**

The license to publish form must be signed before your manuscript can be sent to production. A link to the electronic license to publish form will be sent to the corresponding author only. Please take a moment to check your funder requirements.

Sincerely,

Reviewer #1 (Comments to the Authors (Required)):

I am satisfied with the revisions and recommend to accept the manuscript for publication.

Reviewer #2 (Comments to the Authors (Required)):

The revised manuscript addressed all my concerns and I suggest the editor to accept and publish this excellent manuscript.

Reviewer #3 (Comments to the Authors (Required)):

The authors have compiled an extensive rebuttal letter and made a number of changes in the manuscript to clarify and improve the descriptions of the workflow, the outcomes and the caveats. These changes should greatly help general readers and those interested in the ribotag technology and the genetic prion diseases models.

September 12, 2022

RE: Life Science Alliance Manuscript #LSA-2022-01530-TRR

Dr. Walker Scot Jackson
Linköping University Hospital
Biomedical and clinical sciences
Cellbiologen Plan10
Linköping 58185
Sweden

Dear Dr. Jackson,

Thank you for submitting your Research Article entitled "Translatome profiling in fatal familial insomnia implicates TOR signaling in somatostatin neurons". It is a pleasure to let you know that your manuscript is now accepted for publication in Life Science Alliance. Congratulations on this interesting work.

DISTRIBUTION OF MATERIALS:

Again, congratulations on a very nice paper. I hope you found the review process to be constructive and are pleased with how the manuscript was handled editorially. We look forward to future exciting submissions from your lab.

Sincerely,
